# Myeloid cells interact with a subset of thyrocytes to promote their migration and follicle formation through NF-κB

Rui-Meng Yang[1,10], Shi-Yang Song[1,10], Feng-Yao Wu[1,10], Rui-Feng Yang [2,10], Yan-Ting Shen [3,10], Ping-Hui Tu[1], Zheng Wang[1], Jun-Xiu Zhang[4], Feng Cheng[5], Guan-Qi Gao[6], Jun Liang[7], Miao-Miao Guo[1], Liu Yang[1], Yi Zhou[8,9], Shuang-Xia Zhao [1]✉, Ming Zhan[3]✉ & Huai-Dong Song [1]✉

The pathogenesis of thyroid dysgenesis (TD) is not well understood. Here, using a combination of single-cell RNA and spatial transcriptome sequencing, we identify a subgroup of NF-κB-activated thyrocytes located at the center of thyroid tissues in postnatal mice, which maintained a partially mesenchymal phenotype. These cells actively protruded out of the thyroid primordium and generated new follicles in zebrafish embryos through continuous tracing. Suppressing NF-κB signaling affected thyrocyte migration and follicle formation, leading to a TD-like phenotype in both mice and zebrafish. Interestingly, during thyroid folliculogenesis, myeloid cells played a crucial role in promoting thyrocyte migration by maintaining close contact and secreting TNF-α. We found that *cebpa* mutant zebrafish, in which all myeloid cells were depleted, exhibited thyrocyte migration defects. Taken together, our results suggest that myeloid-derived TNF-α-induced NF-κB activation plays a critical role in promoting the migration of vertebrate thyrocytes for follicle generation.

Congenital hypothyroidism (CH) is one of the most common genetic endocrine disorders, affecting approximately 1 in 3000–4000 newborns worldwide[1–3]. If left untreated, CH leads to irreversible brain dysfunction and dwarfism, known as cretinism. Most CH cases are caused by thyroid dysgenesis (TD), which results from defects in thyroid development[4,5]. However, the molecular mechanisms underlying TD, especially in the maturation of thyroid follicles, remain largely unknown.

Thyroid development is a complex process that is largely conserved across different species. It involves the specification of endodermal precursor cells, thyroid bud formation, relocalization of the thyroid primordium (TP), and maturation into functional follicles[6–11]. Numerous molecular players, including transcription factors Nkx2-1, Foxe1, Pax8, and Hhex, have been found to be associated with early thyroid development. In the E9.5 mouse embryo (about 24 h post-fertilization (hpf) in zebrafish), the thyroid placode is specified by these transcription factors and buds from the anterior foregut endoderm[8,12,13]. The evaginated TP migrates as a coherent cluster under the regulation of signals such as Foxe1 and finally relocalizes to the inferior neck at E13.5 in mouse (about 45 hpf

[1]Department of Molecular Diagnostics & Endocrinology, Shanghai Ninth People's Hospital, Shanghai Jiao Tong University School of Medicine, Shanghai, China. [2]Department of Oncology, Shanghai Medical College, Fudan University, Shanghai, China. [3]Department of Urology, Shanghai Ninth People's Hospital, Shanghai Jiao Tong University School of Medicine, Shanghai, China. [4]Department of Endocrinology, Maternal and Child Health Institute of Bozhou, Bozhou, China. [5]Department of Laboratory Medicine, Fujian Children's Hospital, Fujian Medical University, Fuzhou, Fujian Province, China. [6]Department of Endocrinology, The Linyi People's Hospital, Linyi, Shandong Province, China. [7]Department of Endocrinology, The Central Hospital of Xuzhou Affiliated to Xuzhou Medical College, Xuzhou, China. [8]Stem Cell Program, Boston Children's Hospital and Harvard Stem Cell Institute, Boston, MA, USA. [9]Division of Hematology/Oncology, Boston Children's Hospital and Dana Farber Cancer Institute, Boston, MA, USA. [10]These authors contributed equally: Rui-Meng Yang, Shi-Yang Song, Feng-Yao Wu, Rui-Feng Yang, Yan-Ting Shen. ✉e-mail: zhaozhao1215@126.com; zhanming@shsmu.edu.cn; huaidong_s1966@163.com

in zebrafish[8,14]. After relocalization, the solid TP then gradually transforms into arborized tissue, which facilitates folliculogenesis in all parts of the gland[15,16]. However, the precise mechanisms controlling the transformation of the multilayered TP into a multitude of single-layered follicles remain largely unknown. Moreover, the entire process of thyroid gland development is under the influence of both cell-autonomous and non-autonomous signals, such as those from the cardiac mesoderm and endothelial cells[17–22]. The latest advances in single-cell sequencing and spatial transcriptomic technology provide valuable tools for investigating the cell-cell interactions involved in thyroid folliculogenesis.

In this study, we used in vivo continuous observations of zebrafish thyroid development and multimodal analysis of the transcriptomic heterogeneity of mouse thyroid tissues to identify a subtype of thyrocytes with high motility that drives thyroid folliculogenesis. We found that this process is mediated by TNF-α-induced NF-κB activation, which is triggered by mobile myeloid cells in thyroid tissues.

## Results

### Heterogenous thyrocytes existed during thyroid folliculogenesis

After relocalization to the final position, how the solid TP transforms into organ containing mature follicles, remain largely unknown. By dynamically observation of the process of thyroid follicles formation from TP in zebrafish transgenic lines (*tg:mCherry*) that specifically labeled thyrocytes[6,23], we found that the initial protruding pseudofoot preceded the migration of several thyrocytes (marked with arrowheads), which lasted approximately 7–8 h and then took shape into new follicles (correspondingly numbered and circled with dotted lines) (Fig. 1a). At 80 h post fertilization (hpf) in zebrafish, thyrocytes extending out of the TP reduced cell adhesion as shown by lower E-cadherin expression (Fig. S1a).

In mice, from E15.5, the solid cords gradually transformed into rows of micro-follicles after fusion with the ultimobranchial bodies[7]. We continuously examined the histological changes in mice thyroid tissues from 1 day to 1 year after birth and found that folliculogenesis is not synchronous because peripheral thyrocytes formed a monolayer earlier than the central located thyrocytes, which initially appeared as closed spheres (Fig. 1b). The number of follicles within the mouse thyroid tissues gradually increased until 1 month after birth, with a decreasing number of thyrocytes sticking together aside the follicles (Fig. 1b, c). The size of thyroid lumen otherwise continuously increased in mouse up to 1 year of age (Fig. 1b, c). Thyroid peroxidase (*TPO*), which encodes a membrane-bound glycoprotein, targets the apical membrane of polarized thyroid epithelial cells and participates in the generation of thyroid hormones. In postnatal day 5 mice, the majority of TPO-positive thyrocytes were not arranged into follicles and remained in a primitive state in the thyroid tissues, especially those localized in the center (Fig. S1b, c). Furthermore, in sharp contrast with orderly arranged follicles observed in 1-month-old mice thyroid tissues, abundant thyroid-specific transcription factor 1 (NKX2-1)-positive thyrocytes were disorganized and clustered together at postnatal day 5 (Fig. 1d, e). The number of TPO-positive and thyroxine-containing follicles were gradually increased, suggesting functional maturation accompanies mouse growth (Fig. 1f, g).

Polarized epithelial cells were characterized by compartmentalized expression of adhesion molecules, such as E-cadherin and β-catenin. In the thyroid gland on postnatal day 5, a considerable proportion of thyrocytes had low membrane levels of E-cadherin or β-catenin (Fig. 1h–k and Fig. S1d, e). Correspondingly, the number of thyrocytes expressing E-cadherin or β-catenin on the basolateral membrane increased from postnatal day 5 to 1 month (Fig. 1h–k and Fig. S1d, e), suggesting the presence of heterogeneous thyrocytes during thyroid folliculogenesis.

### A subtype of thyrocytes with higher migratory capacity and TNF-α-NF-κB activation defined by scRNA-seq

To fully comprehend the thyrocyte heterogeneity, the mice thyroid tissues on different postnatal days (5, 10, 20, and 30 days) were dissected to perform single-cell RNA sequencing (scRNA-seq) (Fig. S2a). After standard data processing and quality control procedures (Methods), we obtained transcriptomic profiles for 29,914 cells (Fig. S2b–g). Further integration and unsupervised clustering revealed twelve major cell types in mice thyroid tissues (Fig. S2h). Annotation using canonical markers (Methods) in these 12 clusters identified them as T lymphocyte (T), macrophage (M), B lymphocyte (B), dendritic cell, neutrophil (Neu), mastocyte (Mas), endothelial cell (EC), fibroblast (Fib), epithelial cell (Epi), stromal cell (Str), and lymphatic endothelial cell (LEC), as well as one group of nerve cells (Ner) (Fig. S2h, i). Consistent clustering was obtained by batch effect correction by Harmony method[24], except LEC, which was not separated from EC (Fig. S2k).

The Epi cluster were found including three subclusters: parathyroid cell (PTC), parafollicular cell (C cell) and thyroid follicular cell (TFC) (Fig. S3a–d). Among these, TFC were then selected for in-depth analysis. Supporting the gradual differentiation of TFC during this time window, the expression levels of transcription factors involved in TFC specification, such as *Pax8*, *Nkx2-1* and *Foxe1* were reduced, but the functions related genes such as *Duox2*, *Tg* and *Tpo* were increased in TFC from day 5 to 1 month after birth[6–11] (Fig. 2a). We also found that with thyroid development, the expression levels of genes related to cell adhesion gradually increased, while those related to cell mobility, such as *Vim*, *Mcam* and *Map4k4*, were gradually decreased[25–30] (Fig. 2b). This was also confirmed by immunofluorescence and RNAscope analysis (Fig. 2c–e).

The TFC could be clustered into two subtypes, TFC-1 and TFC-2 by unsupervised clustering (Fig. 2f). With growth, the relative proportion of TFC-1 was reduced, with a corresponding increased fraction of TFC-2 subtype (Fig. 2g). Interestingly, the genes related to positive regulation of cell migration were enriched in TFC-1 when compared with TFC-2 by gene ontology (GO) analysis (Fig. 2h and Fig. S3e). We further found that TNF-α-NF-κB signaling pathway was significantly enriched in TFC-1 cells compared to the TFC-2 subtype according to the results of GO_KEGG signaling pathway analysis and gene set enrichment analysis (GSEA) (Fig. 2i, j and Fig. S3f). This indicates the preferential activation of TNF-α-NF-κB signaling and cell migration pathways in TFC-1 cells. Moreover, a gradual reduction of expression levels of TNF-α-NF-κB transcripts in TFC from postnatal 5 to 30 days were observed (Fig. 2k).

To verify the molecular changes of thyrocytes transition during thyroid development, we performed pseudo-time analysis using Monocle2[31]. Along the pseudo-time trajectory, a gradually reduced faction of TFC-1 and increased TFC-2 from postnatal 5d to 30d were observed (Fig. S4a–c). The result showed that the TFC-1 cells preferentially exist in the early stages of development, and its proportion gradually decreases over time, while the TFC-2 cells increased in the maturation stage of the thyroid. Genes increased in the TFC-1 branch (module 2) were largely involved in TNF-α-NF-κB signaling and EMT transition, while those upregulated in TFC-2 (module 1) were enriched in oxidative phosphorylation (Fig. S4d–e).

### 10 × ST-seq revealed the enrichment of TNF-α-NF-κB activation thyrocytes in the central gland

Building upon our initial observations, we noted the presence of two distinct types of thyrocytes within the mouse thyroid tissue. One type already formed thyroid follicles primarily located in the peripheral region, while the other type, situated in the central gland, had not yet formed follicles. Additionally, our scRNA-seq analysis unveiled the existence of a specific subtype of thyrocytes exhibiting higher migratory capabilities and activation of the NF-κB signaling pathway. In order to get deeper understanding into the relationship

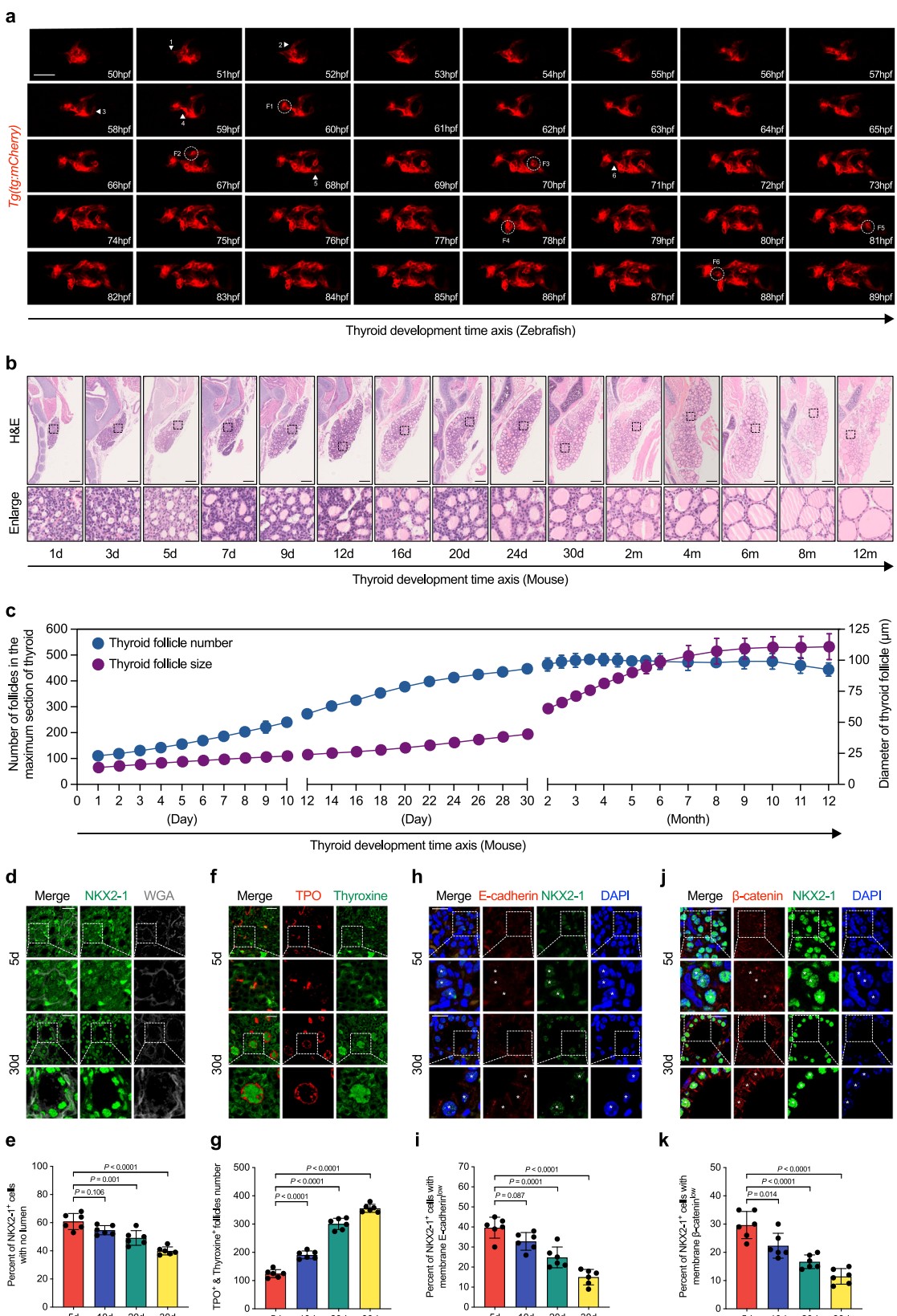

between thyrocyte maturation statuses and their molecular characteristics, we extended our investigation by employing 10× spatial transcriptome sequencing (ST-seq) on thyroid glands obtained from two mice at postnatal day 10. This approach was utilized to elucidate potential associations between distinct maturation states of thyrocytes and various molecular traits (Fig. S5a–j). After data

processing and quality control procedures (Methods), the spots were unsupervised divided into three clusters (TFC, PTCs and Adipo) based on expression of canonical marker genes, which were consistent with distinct histological features shown by hematoxylin and eosin (H&E) staining (Fig. 3a and Fig. S5k). To analyze all pairs of cell type- and thyroid region-specific genes, we integrated the

**Fig. 1 | Gradual transformation of solid primordium into arborized follicles in mice and zebrafish. a** In vivo continuous observation thyroid folliculogenesis of zebrafish embryos from 50 hpf to 89 hpf with one hour interval under *thyroglobulin* promoter driven mCherry transgenic line (*tg*:mCherry). **b** Representative images showing histological analysis at the maximum sections of mice thyroid tissues by H&E staining from postnatal day 1 to 12 months with age shown in the lower panel. **c** Thyroid follicle numbers (y axis on the left) and average follicle diameter (y axis on the right) calculated at the maximum sections of thyroid tissues as time indicated (x axis). **d** Representative images showing the arrangement of NKX2-1 positive thyroid epithelial cells at postnatal day 5 and 30 mice thyroid glands by immunofluorescence analysis (IF). WGA used to stain the membrane. **e** Statistical estimation of the fraction of NKX2-1 positive thyrocytes without lumen formation. **f** Representative images showing the expression of TPO and Thyroxine at postnatal day 5 and 30 mice thyroid glands by IF. Many TPO positive thyrocytes that had yet

to form follicles were found to be thyroxine negative in postnatal day 5 thyroid tissues when compared to postnatal day 30 mice thyroid tissues. **g** Statistical estimation of the percentage of TPO positive lumen with thyroxine secreted. **h–k** Representative images showing the expression of E-cadherin (**h**) or β-catenin (**j**) among NKX2-1 positive thyrocytes at postnatal day 5 and 30 mice thyroid glands by IF. NKX2-1 positive thyrocytes marked with asterisks in postnatal day 5 were membrane E-cadherin (**h**) or β-catenin (**j**) low, in comparison with the clear establishment of epithelial adhesion in postnatal day 30 (also marked with white asterisks). Statistical assessment of the percentage of NKX2-1 positive thyrocytes with low membrane E-cadherin (**i**) or β-catenin (**k**) expression. Scale bar, 50 μm in (**a**, **d**, **f**, **h**, **j**), and 200 μm in (**b**). *n* = 6 biologically independent samples in (**c**, **e**, **g**, **i**), and (**k**). Data are shown as mean ± SD. Statistical significance was determined by One-way ANOVA, followed by Tukey's multiple comparison test. Source data are provided as a Source data file.

scRNA-seq and ST datasets using the traditional multimodal intersection analysis (MIA)[32] (Fig. 3b). Despite the low resolution of ST-seq, which makes it difficult to distinguish single cells scattered throughout the thyroid tissue, we observed significant overlaps between thyroid region-specific and cell type-specific genes, indicating the usefulness of employing MIA to infer cell types identified using scRNA-seq data on the thyroid region by integrating ST data. Interestingly, the two subtypes of TFC identified by further unsupervised spots clustering perfectly matched their locations in the central and peripheral regions of the thyroid tissues shown by HE staining respectively (Fig. 3c). We thus named these two subtypes of spots as TFC-central and TFC-peripheral (Fig. 3c). MIA analysis of TFC subclusters identified using scRNA-seq and ST revealed that TFC-1 subclusters significantly overlapped with TFC-central cells identified using ST data, while TFC-2 subclusters closely overlapped with TFC-peripheral cells (Fig. 3c). Furthermore, based on ST-seq data, genes associated with cell locomotion and the TNF-α-NF-κB signaling pathway were enriched in the TFC-central population (Figs. 3d–g and S5l).

### Thyrocyte subtypes with increased migratory ability were shown with NF-κB signaling activation

We then investigated the activation of NF-κB signaling in thyrocytes and its relationship with maturation and motility. Our results showed that NF-κB pathway transcriptional factor P65 was highly expressed in a subset of thyrocytes, and the percentage of P65 expression cells were gradually decreased as mouse grew older (Fig. 4a, b and Fig. S6a, b). From postnatal day 5 to 30, the prevalence of phosphorylated IKK/IKK or nuclear-phosphorylated P65 cells showed the similar trend (Fig. S6c–f). In the peripheral thyroid glands, where mature follicles had formed with TPO expression on the apical and E-cadherin on the basolateral surface (Fig. 4c, d), P65-positive cells were rarely seen. P65-positive cells, on the other hand, were widely distributed in the central thyroid gland, which contained immature thyrocytes (Fig. 4c, d). Additionally, we found that thyrocytes that were highly P65-positive had a partly mesenchymal phenotype (Fig. 4e, f), which was marked by a high level of motility-related genes like MCAM and Vimentin expression.

We then used double transgenic zebrafish line Tg(NF-κB:eGFP; *tg*:mCherry), where the expression of the green fluorescent protein is triggered by the upstream κB binding sites, to further examine the biological activity of thyrocytes activated by NF-κB[33,34]. We found that NF-κB was activated in a subgroup of thyrocytes with a protruding extension morphology that were in a mesenchymal migration state starting at 2.5 dpf when the TP was situated on the cardiac outflow tract (Fig. 4g). By continuously observing in vivo during the separation of numerous TFC from the current primordium in order to produce new follicles, the NF-κB were found activated in both migration "front" and "rear" thyrocytes (Fig. S7).

### Suppression of NF-κB signaling compromised epithelial motility and thyroid folliculogenesis

To investigate the impact of NF-κB activation on thyroid development, we administered TPCA1, an IKK2 inhibitor, to mice via intraperitoneal injection from postnatal days 3 to 20, at a dosage of 10 mg/kg every 6 days. TPCA1 treatment led to obvious growth retardation, with reduced body size and weight loss (Fig. 5a, b). Moreover, the numbers of strongly P65-positive thyrocytes in the thyroid tissues of mice at postnatal day 30 were significantly reduced compared with those in control mice (Fig. S8a, b). We also observed decreased numbers of thyrocytes expressing migratory genes in the thyroid tissues of TPCA1-treated mice (Fig. S8c–h). After TPCA1 treatment, thyroid function was compromised, as evidenced by the reduced levels of T3 and T4, but increased TSH levels in mice at postnatal day 30 (Fig. 4c–e). Follicle numbers and T4-producing follicle numbers were notably reduced in TPCA1-treated mice at 1 month (Fig. 4f–i). Additionally, tightly-connected epithelial clusters without lumen formation increased in thyroid tissues of mice after TPCA1 treatment, which is consistent with the assumed role of NF-κB activation in promoting epithelial migration and new follicle generation (Fig. 4j, k).

Treatment of zebrafish with TPCA1 also resulted in defective migration of thyrocytes out of the TP, as observed through continuous in vivo monitoring (Fig. S9a). Additionally, T4-producing follicles were reduced in 8dpf zebrafish embryos treated with TPCA1 (Fig. S9b). In zebrafish embryos treated with TPCA1, the number of TUNEL-positive thyrocytes per thyroid gland were slightly increased, suggesting that improper migration may eventually cause these cells to undergo apoptosis (Fig. S9c). BMS-345541, another selective inhibitor of the catalytic subunits IKK1 and IKK2, was also used to examine follicle formation in zebrafish embryos from 24 hpf. Similar follicle formation defects were observed in BMS-345541 treated embryos compared to controls (Fig. S9d). Furthermore, we specifically overexpressed the NFKBIA (IκBα) in thyrocytes under the thyroglobulin promoter, which retains NF-κB members in the cytosol to resist NF-κB activation[35]. Transient overexpression of this plasmid in thyrocytes induced similar thyroid abnormalities, confirming the intrinsic role of NF-κB signaling in thyroid development (Fig. S9e). To discriminate of the role of NF-κB signaling on the early whole thyroid gland migration process, we examined *tg* expression in 50hpf zebrafish by whole-mount in situ hybridization (WISH). Neither TPCA1 nor BMS-345541 treatment affected TP relocalization to the heart outflow shown in 50hpf zebrafish (Fig. S9f–g).

Our findings show that defective follicle formation was also observed in zebrafish embryos treated with a selective Rac1 inhibitor (Rac1 is a key protein that promotes cell migration), NSC 23766 trihydrochloride, supporting the importance of thyrocyte migration in thyroid development (Fig. S9h). Moreover, when Rac1 was overexpressed in the thyrocytes of zebrafish embryos, the defects in thyroid development caused by NF-κB suppression were largely

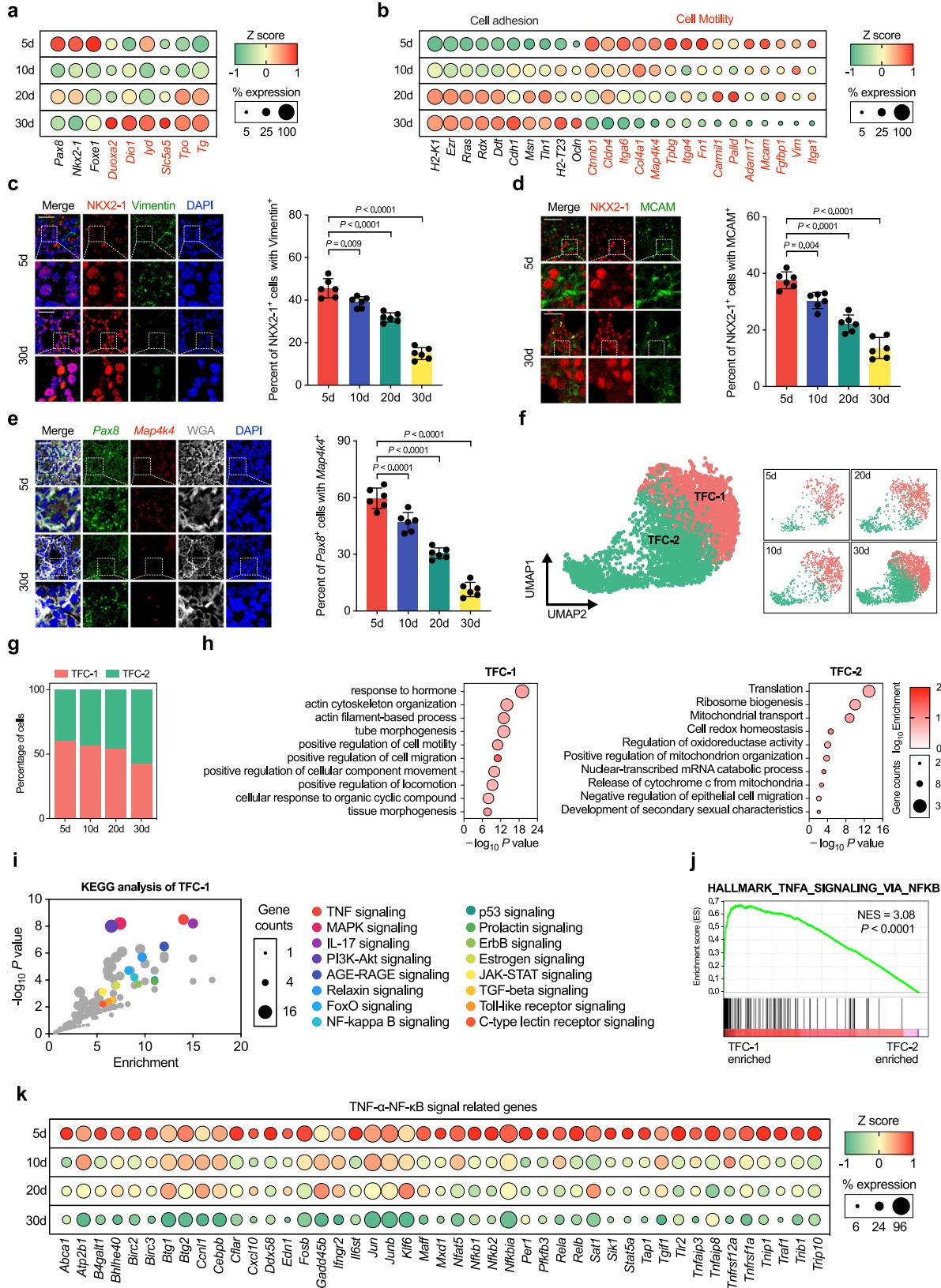

restored, validating that thyrocyte migration is a critical downstream effector of NF-κB activation (Fig. S9i).

**Myeloid cells in the developing mouse and zebrafish thyroid tissue secrete TNF-α to trigger the NF-κB activation of TFC**

To investigate how different cell types interact with TFC during mouse thyroid development, we computed interaction scores based on the

average expression levels of a ligand and its receptor between cell subtypes in our scRNA-seq data using CellPhoneDB (Fig. S10a)[36,37]. The number of interaction pairs targeting epithelial cells decreased with postnatal age (Fig. S10b). TNF-α-mediated interactions with TFC were mainly derived from CD68- and IL-1β-expressing macrophages, dendritic cells, and neutrophils (Fig. S10c–f). Notably, TNF-α-mediated interactions were enriched in TFC-1 cells, supporting the preferential

**Fig. 2 | Identification of motile thyrocytes by scRNA-seq from the developing mice thyroid tissues. a** Dot plot of thyroid specification (marked in black) and function related genes (marked in red) expression in thyrocytes at each time point. **b** Dot plot of cellular adhesion and motility related genes expressed in thyrocytes at each time point in postnatal mice thyroid tissues (left axis). **c** Representative images and statistical assessment showing the percentage of Vimentin positive thyrocytes (stained by NKX2-1) from postnatal day 5 to 30 mice thyroid tissues. **d** Representative images, and statistical assessment showing the MCAM positive thyrocytes in postnatal day 5 and 30 mice thyroid tissues by using IF staining. **e** Representative images, and statistical assessment of RNA-scope analysis of *map4k4* positive thyrocytes (stained by *Pax8* as a marker for thyrocyte) from postnatal day 5 to 30 mice thyroid tissues. **f** UMAP of TFC and split by time point. Cells are colored and annotated by cell subtype. **g** Bar plot shown the proportion of

TFC subtypes among all TFC at each time point examined. **h** GO enrichment analysis of the DEGs that specifically enriched in TFC-1 and TFC-2 subtype respectively. **i** KEGG analysis of the DEGs distinguishing the TFC-1 subtypes. **j** Analysis of "HALLMARK_TNFA_SIGNALING_VIA_NFKB" gene set between TFC-1 and TFC-2 cells by GSEA software, the NES and FDR *P* value were shown. **k** Dot plot of TNF-α-NF-κB pathway genes expression in TFC cluster at each time point in postnatal mice thyroid tissues. For (**a**, **b** and **k**), color of dots represents z-scored of the gene expression level, and size of dots represents percent of TFC with at least one UMI detected per gene. In (**c**, **d** and **e**), Scale bar, 50 μm. *n* = 6 biologically independent samples. Data are shown as mean ± SD. Statistical significance was determined by One-way ANOVA, followed by Tukey's multiple comparison test. In (**h** and **i**), *P* value was determined by Benjamini-Hochberg-adjusted one-sided hypergeometric test. Source data are provided as a Source data file.

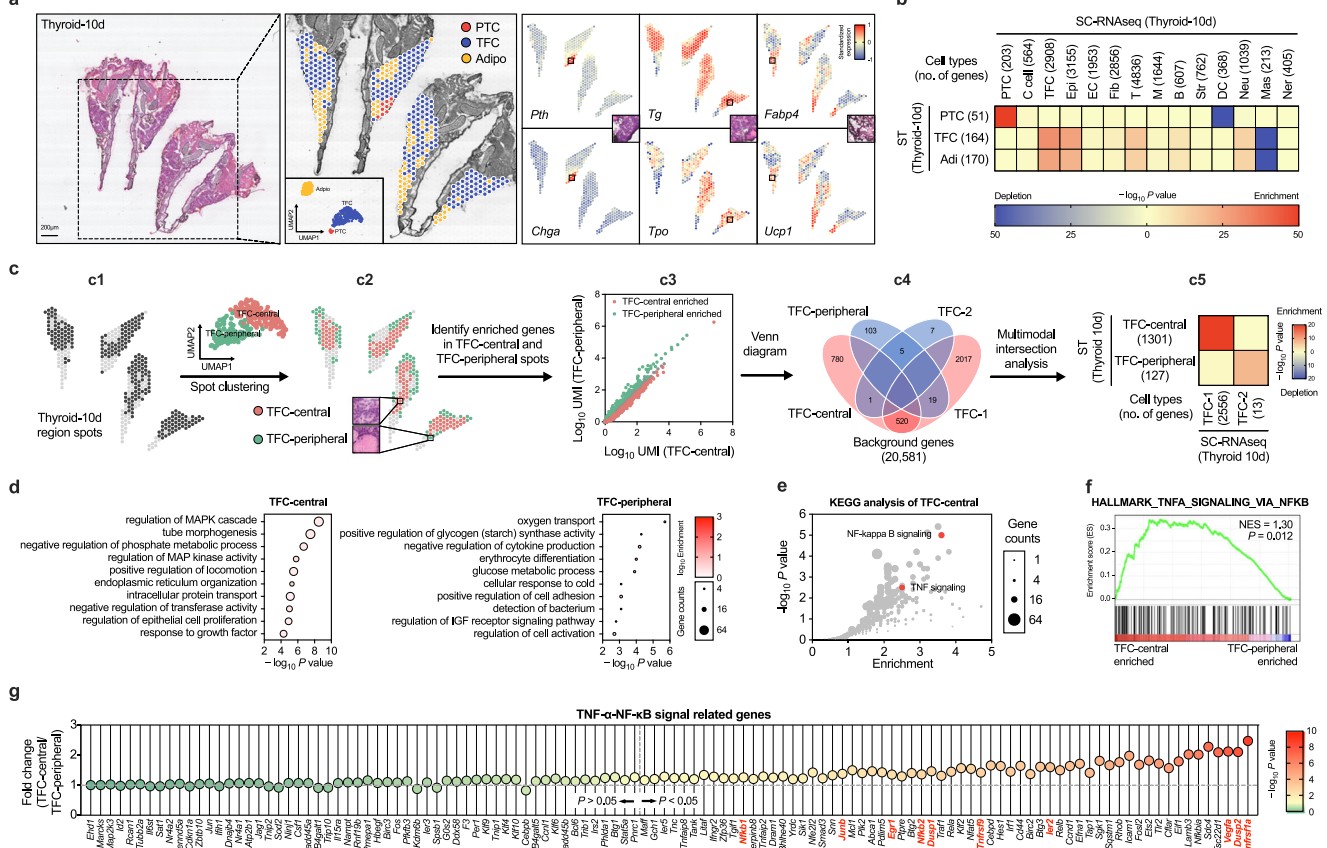

**Fig. 3 | NF-κB is activated in the central immature thyrocytes of mice by spatial transcriptome sequencing. a** From left to right in turn show the H&E staining, cell type annotation and representative marker genes expression in thyroid tissues. PTC: Parathyroid cells (PTC), TFC: thyroid epithelial cells (TFC), Adipo: adipocyte. Dot color intensity represents the z-score of gene expression values. **b** Multimodal intersection analysis (MIA) map of all scRNA-seq-identified cell types and ST-defined regions. *P* value calculated by hypergeometric test. The numbers of cell type- and tissue region-specific genes used in the calculation are shown in the brackets. **c** MIA analysis was utilized to examine the relation between scRNA-seq-identified TFC subtypes and ST-defined TFC subclusters. Based on our ST-seq data, UMAP analysis identified two TFC subclusters (c1), one of which was in the central of thyroid tissues and the other in the peripheral region (c2). Genes with significantly higher expression in each spatial region relative to the others were then identified (c3). The overlap between each pair of cell type-specific and tissue region-specific gene sets was analyzed using MIA (c4). The significance of the

intersection was displayed using the hypergeometric distribution. (c5). The numbers of cell subtype-specific and tissue region-specific genes used in the calculation are shown in the brackets. **d** GO enrichment analysis of the DEGs between TFC-central and TFC-peripheral. **e** KEGG analysis of the TFC-central highly expressed genes. Red dots signify the genes in the TNF and NF-κB signaling pathway. **f** Analysis of "HALLMARK_TNFA_SIGNALING_VIA_NFKB" gene set between TFC-central and TFC-peripheral spots by GSEA software, the NES and FDR *P* value were shown. **g** Fold changes of TNF-α-NF-κB pathway genes comparing TFC-central with TFC-peripheral cell spots. Dot color intensity represents the -log$_{10}$ *P* values (Two-tailed *t* test). The dotted horizontal line indicates fold change = 1, and the dotted vertical line show *P* = 0.05. Gene names marked in red were also highly expressed in TFC-1 subtype when compared with TFC-2 by scRNA-seq. In (**d** and **e**), *P* value was determined by Benjamini-Hochberg-adjusted one-sided hypergeometric test. Source data are provided as a Source data file.

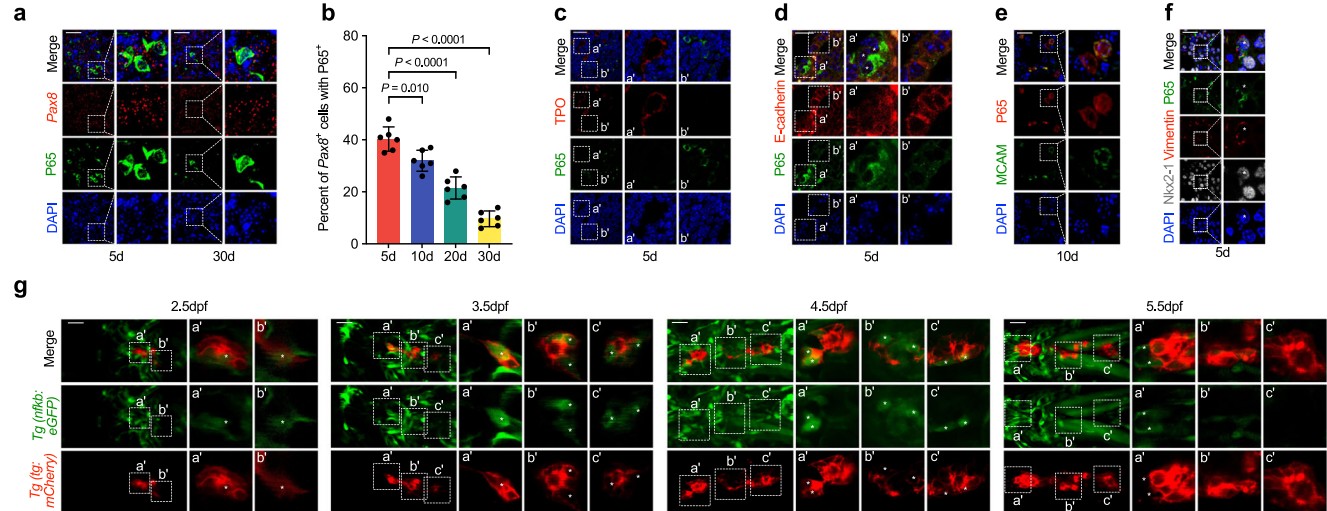

**Fig. 4 | NF-κB signaling is activated in TFC with higher migratory capacity in zebrafish and mice. a, b** Representative images (**a**) and statistical assessment (**b**) of the percentage of P65 positive thyrocytes, which are stained by *Pax8* RNA probe from postnatal day 5 to 30 mice thyroid tissues. **c** P65 co-expression with TPO in postnatal day 5 mice thyroid tissues were analyzed by IF staining. Right two panels are enlarged ones collected from the peripheral (a') and central (b') region of the first column on the left respectively. Note that P65 positive cells were accumulated in the central region (b'), with apical TPO and lumen not established. **d** Cellular adhesion molecule E-cadherin expression in P65 positive cells in postnatal day 5 mice thyroid tissues were detected by IF staining. Right two panels are the enlarged ones of the first lane showing P65 positive cells in the central gland with membrane E-cadherin negative (a') and P65 negative cells in the peripheral part with membrane E-cadherin positive (b') respectively. **e** Representative images showing

MCAM expression in P65 positive cells by IF in postnatal day 10 mice thyroid tissues. **f** Representative images showing Vimentin expression in P65 positive thyrocytes in postnatal day 5 mice thyroid tissue, labeled by white asterisks. **g** NF-κB activation in thyrocytes in zebrafish embryos at different time points were examined by the double transgenic Tg(*nfkb*:eGFP; *tg*:mCherry) line. The first columns in each panel show the whole mount embryos view of thyroid glands for each time point of embryos. The second and third columns are magnified and slice images of the first columns, respectively, to clearly show NF-κB activation in thyroid epithelial cells. Scale bar, 50 μm. Three independent experiments were carried for (**c**–**g**). In (**b**), data are shown as mean ± SD, n = 6 biologically independent samples, and statistical significance was determined by One-way ANOVA, followed by Tukey's multiple comparison test. Source data are provided as a Source data file.

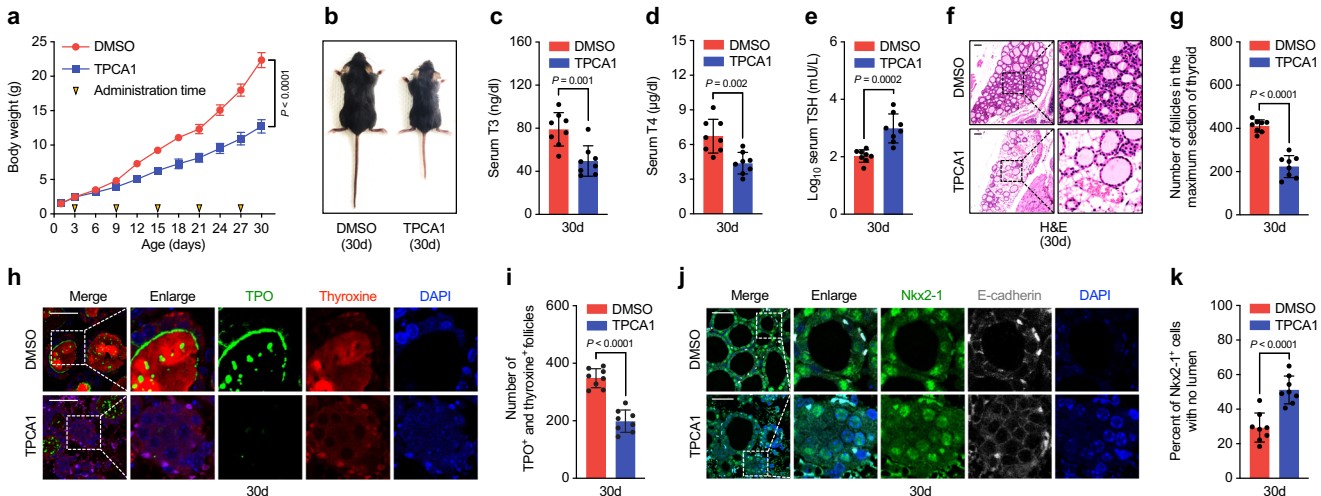

**Fig. 5 | NF-κB inhibition affect thyroid follicle formation in mice. a** The changes of body weight in mice treated by IKK2 inhibitor TPCA1. Yellow inverted triangles in X axis indicated the administration time of TPCA1, and the body weight of mice were measured every three days. Repeated measures ANOVA used for significance testing. **b** Representative image show the body size of mice treated by TPCA1 under stereoscope. The serum levels of T3 (**c**), T4 (**d**) and TSH (**e**) in the mice after treated with TPCA1 for one month. **f** Representative image of histology analysis of thyroid tissues at postnatal day 30 mice after treated with TPCA1. **g** Statistical assessment of follicle numbers formed at the maximum sections of thyroid tissues in mice treated with TPCA1. **h** Representative images showing TPO and Thyroxine levels in thyroid

tissues of the postnatal day 30 mice treated with or without TPCA1. **i** Statistical measurement of TPO and thyroxine positive lumens (indicated of functional maturation) in thyroid tissues of the postnatal day 30 mice treated with TPCA1. **j** The expression of NKX2-1 and E-cadherin in thyroid tissues of postnatal day 30 mice treated with TPCA1 were detected by IF. **k** Statistical assessment of the percentage of thyroid epithelial cells clustering into solid mass (with no lumen formed and no polarized E-cadherin expression) in thyroid tissues of postnatal day 30 mice treated with TPCA1. Scale bar, 50 μm. n = 8 biologically independent samples in (**a**, **c**, **d**, **g**, **i**, **k**). Two-sided Student's t test used. Data are shown as mean ± SD. Source data are provided as a Source data file.

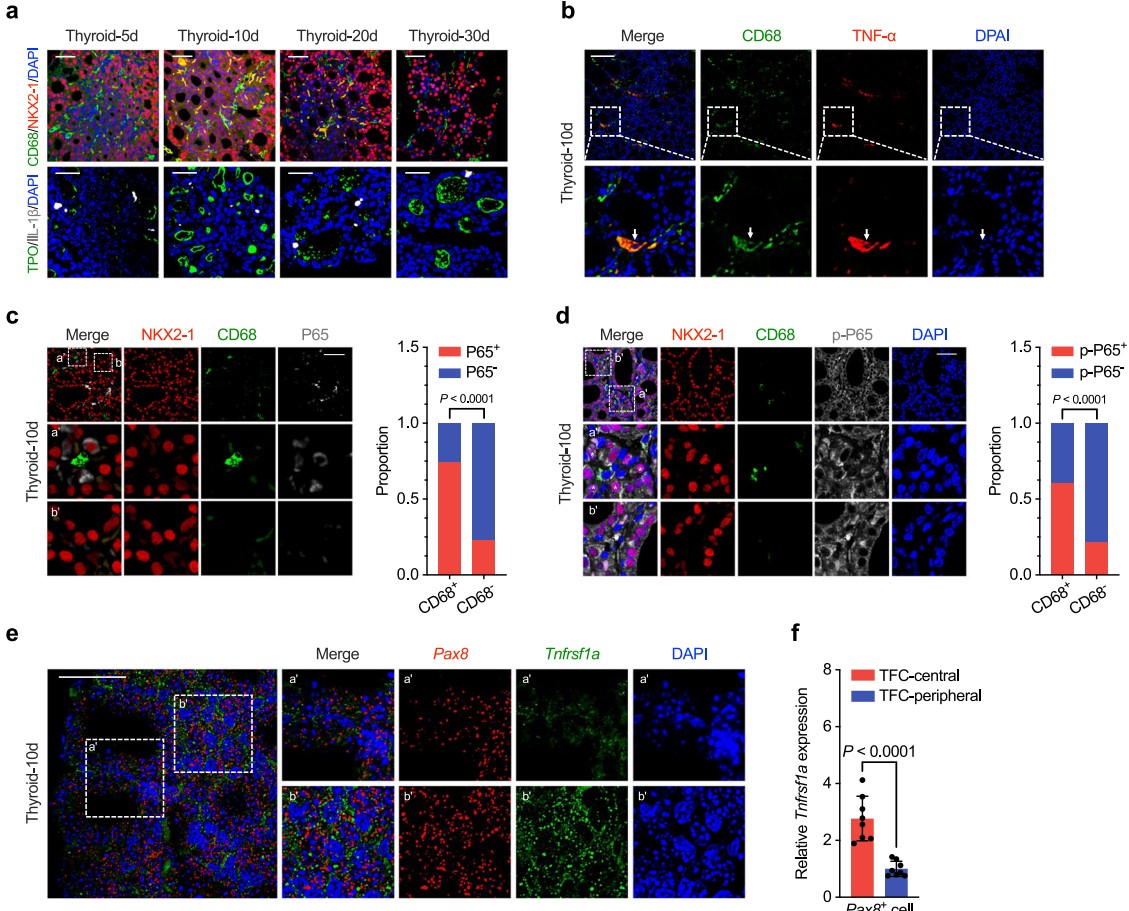

**Fig. 6 | Myeloid cells secrete Tnf-α to promote NF-κB activation in mice thyroid epithelial cells. a** The CD68 and Il-1β positive cells were identified in thyroid tissues of postnatal day 5, 10, 20, 30 mice by IF. **b** Representative images of IF analysis showing Tnf-α expression in CD68 positive macrophages (arrowheads) at postnatal day 10 mice thyroid tissues. Representative image and statistical analysis showing CD68-positive macrophages located near the P65 (**c**) or nuclear phosphorylated P65 (**d**) positive thyrocytes in thyroid tissues at postnatal day 10 mice by poly-chromatic immunofluorescence analysis. **e** Representative image showing the multi-color RNA scope analysis of *Pax8* and *Tnfrsf1a* in postnatal day 10 mouse thyroid gland. The areas framed by the two white dashed lines correspond to the peripheral (a', mature lumen has been established) and central (b', immature solid mass appearance) of thyroid tissue, which are shown on the right. **f** Statistical assessment of the relative fluorescence intensity of *Tnfrsf1a* in the thyrocytes located in the central or peripheral thyroid tissues at postnatal day 10 mice. Of the 6 mice thyroid tissues examined, three fields of view for each were randomly selected for calculation the mean fluorescence intensity (divided by the number of cells) of *Tnfrsf1a* in the peripheral and central thyrocytes, respectively. Data are shown as mean ± SD, and and statistical significance was determined by Two-sided Student's *t* test (**f**). Scale bar, 50 μm. Three independent experiments were carried for (**a** and **b**). For the statistical analysis in (**c** and **d**), 70 areas were randomly chosen under confocal microscopy from the slices of 7 postnatal day 10 mice thyroid tissues (10 areas for each), the areas were classified into two groups by whether CD68 positive cells existed. The presence of P65 (**c**) or nuclear phosphorylated P65 (**d**) positive cells or not in each group were summed respectively. Two-sided Fisher's exact test to calculate the *P* value. Source data are provided as a Source data file.

activation of the NF-κB signaling pathway in these cells (Fig. S10g). Using immunofluorescence, we confirmed the presence of CD68-positive macrophages, as well as the presence of IL-1β-stained neutrophils, in postnatal mouse thyroid tissues (Figs. 6a and S11a). We also validated TNF-α expression in CD68-positive macrophages using IF assay (Fig. 6b). Thyrocytes positive for P65 or nuclear phosphorylated P65 were found in close proximity to CD68-positive macrophages (Fig. 6c, d). Furthermore, we identified *Tnfsf1ra*, a receptor for TNF-α, in TFC using scRNA-seq and confirmed its expression was higher in TFC-central cells than in TFC-peripheral cells in mouse thyroid tissues using RNA-scope analysis (Fig. 6e, f).

With the use of the zebrafish double transgenic line, we showed that from 2 dpf, the zebrafish TP had been surrounded by both macrophages [Tg(*mpeg*:eGFP)] and neutrophils [Tg(*mpo*:eGFP)] and (Fig. 7a, b). We observed through continuous in vivo studies that myeloid cell–thyrocyte interaction increased thyrocyte migration and epithelial sheet separation in TP of zebrafish embryos (Figs. 7a, b, and S11b). We observed that TNF-α was expressed in neutrophils of the

TP at 4.5 dpf embryos, which is in consistent with the findings in the thyroid tissues of mice (Fig. S11c).

Similar to the abnormal development of TP in zebrafish embryos treated with NF-κB inhibitors, fewer follicles and elevated *tshba* in *cebpa* knockout zebrafish, of which both macrophages and neutrophils were depleted were observed[38] (Fig. 7c and Fig. S12a). The early migration of the whole TP to the cardiac outflow was not affected in 50hpf *cebpa* mutants, suggesting an influential role during later folliculogenesis stage (Fig. S12b). Otherwise apoptotic thyrocytes increased in *cebpa* mutants compared to wild-type zebrafish embryos (Fig. S12c).

To dynamically observe the interaction between TNF-α-expressing cells and thyrocytes, a stable TNF-α reporter transgenic line was generated as previously described[39]. From 2 dpf embryos, the TP was surrounded by TNF-α-positive cells, with morphology resembling macrophages and neutrophils (Fig. 7d). Following interaction with TNF-α-expressing cells, we observed the extension of front thyrocytes (Fig. 7e). Both Tnf transcript levels, the number of TNF-α-

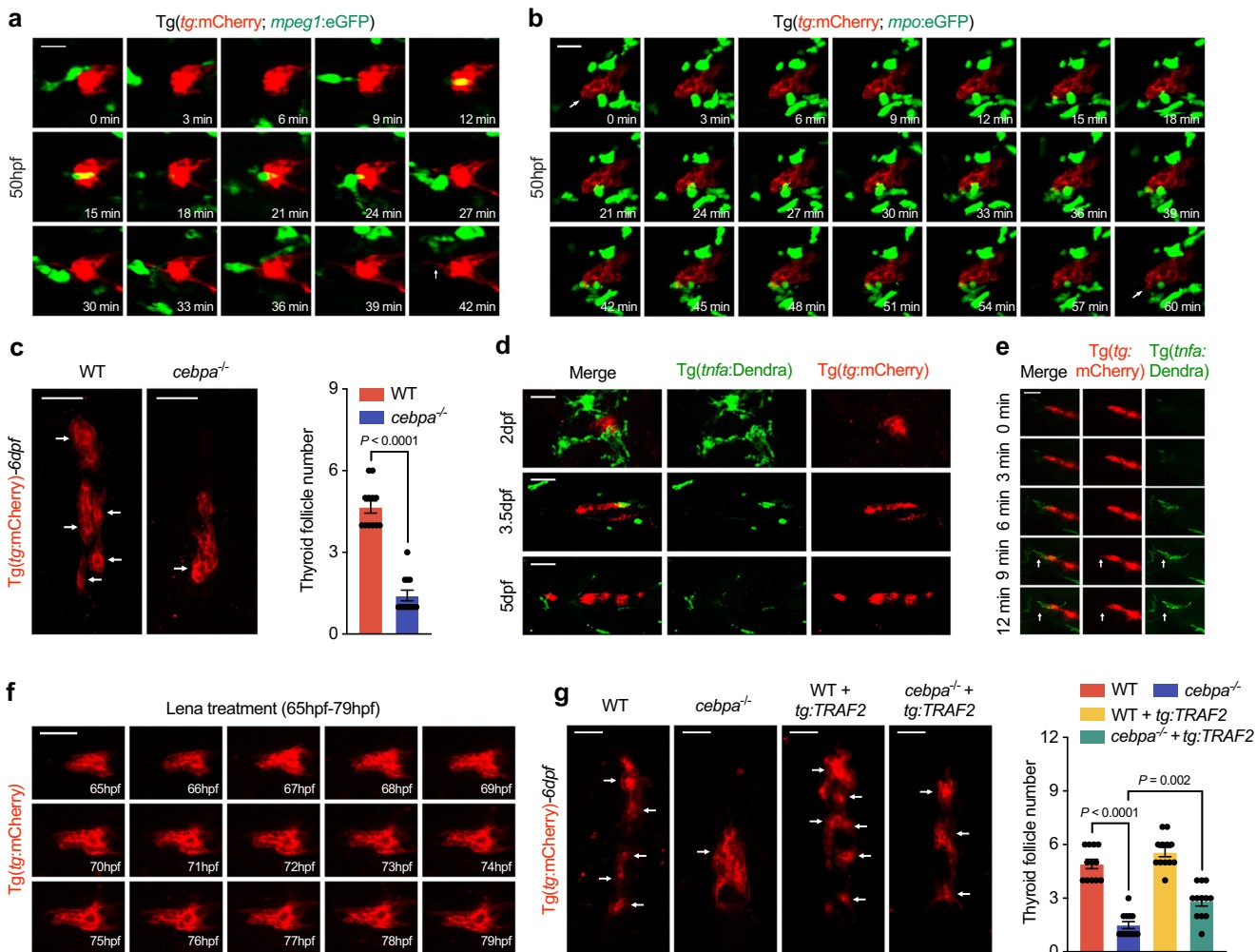

**Fig. 7 | Cells of myeloid lineage secrete Tnf-α to promote NF-κB activation in zebrafish thyroid epithelial cells. a** In vivo continuous observation of the interaction between macrophages [Tg(*mpeg*:eGFP)] and thyroid epithelial cells [Tg(*tg*:mCherry)]. Arrowhead notifies the extension of front epithelial cell after frequent interactions. **b** Interaction between neutrophils [Tg(*mpo*:eGFP)] and thyroid epithelial cells [Tg(*tg*:mCherry)]. Arrowheads indicate the extrusion of front epithelial cell. **c** Representative image and statistical analysis showing follicles formed in 6dpf wild-type and *cebpa*⁻/⁻ under Tg(*tg*:mCherry) background. **d** Tnf-α positive cells [Tg(*tnfa*:Dendra)] in touch with thyroid epithelial cells examined at times indicated on the left in zebrafish embryos. **e** In vivo continuous monitoring

shows the interaction with Tnf-α positive cells promoted the extrusion of front thyroid epithelial cells. **f** Continuous in vivo observation of thyroid epithelial cell morphogenesis in zebrafish embryos treated with lenalidomide. **g** Representative images and statistical analysis of follicle numbers in wild-type and *cebpa*⁻/⁻ with or without TRAF2 overexpression in thyrocytes. Scale bar, 50 μm. Three independent experiments were carried for (**a** and **b**). In (**c** and **g**), data are shown as mean ± SD (*n* = 12 zebrafish/group, significance was determined by two-side Student's *t* test or one-way ANOVA with post Tukey's multiple comparisons test). Source data are provided as a Source data file.

positive cells surrounding the development of TP were significantly reduced in *cebpa* mutants (Fig. S12d, e). The number of NF-κB-activated TFC were significantly reduced in *cebpa* mutants and lenalidomide- (CC-5013), a TNF-α secretion inhibitor[40,41], treated embryos, supported the notion that myeloid cells secrete TNF-α to activate the NF-κB pathway in thyrocytes (Fig. S12f, g). Moreover, lenalidomide treatment suppressed TFC from migrating out of the TP in zebrafish (Fig. 7f). Similar epithelial cell clumping together were revealed by *slc5a5* staining in 5 dpf lenalidomide treated embryos (Fig. S12h). TNF receptor-associated factor 2 (TRAF2) plays a crucial role in the transmission of NF-κB signaling[42]. Overexpression of TRAF2 in thyrocytes partially restored the folliculogenesis defects in either *cebpa* mutants or lenalidomide-treated embryos (Fig. 7g and Fig. S12i). Rac1 overexpression restored the migration capacity of thyrocytes in the TP and partially restored the number of follicles formed in 5 dpf lenalidomide-treated embryos (Fig. S12j). In zebrafish embryos, the interactions between macrophages and TFC were not influenced with lenalidomide treatment (Fig. S13a). Pharyngeal vessel remodeling and late thyroid

morphogenesis are not only spatio-temporally correlated, but also functionally linked[43]. Therefore, using zebrafish tg(*flk*:eGFP) transgenic line, we studied the development of heart and major vessels surrounding the thyroid gland with TPCA-1, BMS or lenalidomide treatment. At the concentration of drugs used, none of them could influence the development of pharyngeal vessels, suggesting relative specific role of TNF-α-NF-κB signaling pathway in thyroid folliculogenesis (Fig. S13b, c).

Collectively, myeloid lineages, including neutrophils and macrophages, secreted TNF-α, which promoted the activation of NF-κB in the proportion of immature thyroid epithelial cells, which triggered the adoption of the mesenchymal migratory phenotype for TP remolding. In vitro co-culture with the neutrophil cell line HL-60 increased the migration related protein expression in normal thyroid epithelial cell line Nthy-Ori3, and these effects were weakened by lenalidomide treatment (Fig. S13d). BMS treatment did not influence de novo folliculogenesis, and interestingly, TNF-α treatment stimulated the migration of thyrocytes in 3D thyroid follicle generation model using

primary mouse thyrocytes (Fig. S13e). These results suggest that TNF-α-mediated NF-κB might not be required for follicle formation from single cells in vitro but is indispensable for the arborization or fragmentation of solid TP into follicles in vivo.

## Discussion

After cell specification, budding, relocalization, and bifurcation, the thyroid lobe, composed of a stratified solid mass of non-polarized cells, needs further fragmentation or arborization to form a multitude of single-layered mature follicles. Infiltration of vascular cells is required for the fragmentation of thyroid masses. The expression of adhesion molecules is highly dynamic during thyroid folliculogenesis and is a critical parameter for mass fragmentation[15]. Through continuous in vivo observation of the morphogenesis of the TP in zebrafish embryos, we found that finger-like thyrocyte protrusions extended out of the TP, which was followed by the collective migration of several thyroid epithelial cells to form new follicles. It seemed that thyrocytes actively migrated out of the TP but were not solely passively fragmented by vascular cells during thyroid folliculogenesis. After successive rounds of protrusion, migration, separation, and lumenogenesis, rows of thyroid follicles were formed along the midline in zebrafish embryos. Folliculogenesis was not synchronous, as revealed by the histological differences in thyrocytes in the central and peripheral regions of the mouse thyroid gland. Peripheral thyroid epithelial cells form open lumens at an earlier stage than central cells; the latter mostly show closed spheres at birth in mice. Through scRNA-seq and ST-seq, we identified two subtypes of thyroid epithelial cells, TFC-1 and TFC-2, in postnatal mice thyroid tissues at different stages. TFC-1, which is characterized by a mesenchymal phenotype and high migratory gene expression, is mostly localized in the central region of the thyroid gland. EMT or partial EMT of the epithelium plays a pivotal role during pancreatic islet formation[44]. Before the formation of endocrine islets, endocrine precursor cells are required to detach from the trunk epithelium, mediated by EMT[44]. Interestingly, the TFC-1 subtype was characterized by the activation of NF-κB and a high migratory capacity when compared with TFC-2 cells by scRNA-seq. In the present study, either suppression of NF-κB or motility of thyrocytes led to similar follicle growth defects in the mouse and zebrafish models, which could be restored by enhancing thyroid epithelial motility via Rac1 overexpression. Previous studies have reported that NF-κB is a ubiquitous transcription factor that plays a prominent role in mediating cells mesenchymal differentiation[45]. The complexes of NF-κB are activated by two major signal transduction pathways, namely, the canonical NEMO/IKKγ-dependent pathway and the non-canonical NEMO/IKKγ-independent pathway. Interestingly, in NEMO thyroid-specific knockout mice, the thyroid follicular architecture appears highly variable in diameter with irregular outlines, and significant thyroid cell loss accompanied by massive apoptosis has been observed[46]. Thus, all the data indicated that the TFC-1 subtype might play key roles in the formation of thyroid follicles when they obtain their migratory capacity, mediated by the activation of the NF-κB pathway, resulting in the EMT of cells. In fact, TNF-α inhibitory monoclonal antibody treatment of adult rheumatoid arthritis patients did not significantly alter thyroid function, suggesting that this signaling pathway might not be pivotal for proliferation and maintenance of thyroid follicles in adults[47].

Directional cell migration is a complex and highly regulated process that requires constant crosstalk between the cells and their environment[48,49]. Cells must be able to perceive various environmental cues, often in the form of a gradient, and orient themselves such that they can move directionally toward the signal[4]. However, at this point, the morphogenesis of the TP seemed less similar to branching morphogenesis, as observed with other exocrine organs. The migration of thyroid epithelial clusters is not synchronized in the same direction but is more stochastic and local. Using scRNA-seq, we found that TNF-α derived from myeloid lineage cells induced the activation of NF-κB, which might mediated by the expression of TNF-α receptor Tnfsf1ra in TFC. The interaction of different myeloid cells with TFC in zebrafish embryos during the TP development was also confirmed. Interestingly, we provided a mechanism by which these mobile macrophages or neutrophils precisely send the TNF-α migration signal to target cells to induce NF-κB activation. The action model of TNF-α with its receptor in TP differed from the known action model of hormones and cytokines; therefore, we defined this action model of TNF-α in TP as "mobicrine", meaning that when the mobile neutrophils or macrophages invade target tissues, they would carry and release hormones or cytokines into targeted sites, but would be taken away when they detach from the targeted cells. The interaction of myeloid cells with targeted cells were located in specific regions rather than the whole thyroid gland, providing cues for triggering the migration and separation of thyrocytes from the TP. Although the number of myeloid cells remained largely unchanged during various postnatal stages, the expression levels of Tnfrsf1a in thyrocytes were gradually decreased, suggesting that this signaling pathway is tightly regulated and developmentally required for thyroid folliculogenesis. It should be noted that we could not exclude the participation of other pro-inflammatory molecules, such as IL-1β, since they were also highly expressed in myeloid cells and could have activated NF-κB signaling. Future investigations might be needed to clarify this. Another interesting point is that a transient immune/inflammatory interference might also occur during the early development in fetuses and neonates of mothers with autoimmune thyroiditis, and our models here could possibly be consistent with transient forms of hypothyroidism evolving later in colloid goiters. Further studies of thyroid size and function in adult mice at later time points after a short early course of TPCA1 treatment would be helpful to answer this.

## Methods

### Study approval

All studies were conducted according to the guidelines approved by the ethics committee of Shanghai Ninth People's Hospital (Approval No. SH9H-2021-A99-1).

### Animals

Wild-type (WT) mice were obtained from Cyagen Biosciences, China. All mice were housed in a pathogen-free environment with the temperature maintained at $23 \pm 2\,°C$ and relative humidity at 50–65% under 12 h light/dark cycle. Mice were fed with normal chow (cat:19123123, Beijing KeaoXieli Feed Co., Beijing, China) with free access to food and water. Mice were euthanized with a combination of $CO_2$ and cervical dislocation to guarantee the death of the animals. Zebrafish were housed in the zebrafish center of Shanghai Ninth People's Hospital. 5–20 adult mixed-sex fish per liter were grown in water with temperature, pH, and conductivity monitored daily and maintained at 26–29 °C, 7–8, and 200–3000 micro-Siemens, respectively. Live brine shrimp in addition to standard diet were used to feed adult fish.

### 10 × single-cell RNA sequencing (scRNA-seq) of mice thyroid tissues

**Single-cell separation of mice thyroid tissues.** Thyroid tissues dissected from C57BL/6 mice of postnatal days 5, 10, 20, and 30 were washed twice in 1 × PBS (Gibco) and cut into 1 mm³ pieces. Tissue pieces were then enzymatically digested with a 3 mL digestion medium containing 5 mg/mL collagenase I/II, 0.2 u/mL dispase (Gibco), and 0.1 mg/mL DNAase. These samples were subsequently incubated at 37 °C for 30 min in a water bath shaker. The suspended cells were washed with 20% fetal bovine serum (FBS) in Dulbecco's modified Eagle's medium (DMEM), filtered through a 40 μm Cell-Strainer nylon mesh (BD), and centrifuged at 700 × g for 10 min. After washing with

MACS buffer, dead cells were removed using the Dead Cell Removal Kit (Miltenyi Biotec, Bielefeld, Germany). The samples with cell viability >90% were assessed by trypan blue staining, were passed for initial quality control.

**Single-cell RNA sequencing.** scRNA-seq was performed using the 10 × Genomics platform, and single-cell suspensions were prepared as described above. Sample libraries were prepared according to the Chromium Single Cell 3′ Library and Gel Bead Kit instructions (v2-v3) and were sequenced on an Illumina NovaSeq 6000 System. The gene-barcode matrices were generated using CellRanger's (v2-4) recommended pipeline, which aligned the droplet-based sequencing data against the mouse mm10 reference genome and counted the UMIs for each cell. The output was a count matrix containing all UMI counts for each droplet.

**Data processing and Quality control of scRNA-seq data.** Sequences obtained using the 10 × Genomics scRNA-seq platform were demultiplexed and mapped to the mouse mm10 reference genome using CellRanger (v3.0.1) recommended pipeline. The output of Cell Ranger is a raw count matrix for each sample, which records the unique molecular identifier (UMI) counts per gene and associated cell barcodes. Low-quality cells and genes were removed using Seurat V3.6.3[50] (http://satijalab.org/seurat/). We first filtered out low-quality cells that fit any of the following criteria: proportion of mitochondrial gene counts >10%, UMIs <200, or UMIs >10,000. Cells expressing high levels of hemoglobin-encoding genes were also discarded. Genes expressed in less than three cells were removed. Computational inference of doublets was performed by the Scrublet package[51]. The expected doublet rate was set to 0.06. Doublets identified in each sample individually were excluded from the following analyses. After excluding low-quality cells and potential duplicates, we obtained 29,914 cells in total: 6481 cells from 10 mice on postnatal day 5; 5110 cells from 10 mice on postnatal day 10; 5996 cells from 10 mice on postnatal day 20; and 12,327 cells from 10 mice on postnatal day 30.

**scRNA-seq data analysis.** Data normalization, dimensional reduction, batch effect removal, clustering, and visualization were performed with the Seurat package. The four datasets were integrated using the SCTransform integration workflow on Seurat. The FindNeighbors() and FindClusters() function in Seurat were used for cell clustering. Meanwhile, shared nearest neighbor graph-based clustering was performed on the principal components analysis (PCA)-reduced data to identify the cell clusters. The resolution was set to 0.9 to obtain the major cell types of thyroid tissues. Seurat's DimPlot function was used to generate the UMAP (Uniform Manifold Approximation and Projection). The cell identity of each cluster was determined with SingleR (v1.0.6) package[52] and then refined based on the expression of canonical marker genes. We identified endothelial cells (*Flt1, Kdr, Podxl*), fibroblasts (*Dcn, Col1a1, Col1a2*), macrophages (*C1qa, C1qb, Cd68*), neutrophils (*S100a8, Il1b*), epithelial cells (*Epcam, Chchd10, Tg*), T cells (*Cd3d, Cd3g, Trbc1, Trbc2*), B cells (*Cd79a, Cd79b, Igkc*), stromal cells (*Acta2, Tagln, Myh11*), dendric cells (*H2-Ab1, H2-Aa, Plbd1*), lymphatic endothelial cells (*Lyve1*), mast cells (*Il7r, Gata3,* and *Rora*), and nerve cells (*Plp1*), which represented all major cell types in thyroid tissues. The Harmony method was also used for Batch effects correction to confirm clustering consistency[24].

**Differential expression and functional enrichment analysis.** To identify marker genes for each cluster, the FindAllMarkers() function using default parameters, which implements a Wilcoxon rank-sum test comparing the gene expression of cells within a given cluster versus all other cells, was used. This test was also used for identifying the marker genes for TFC-1 and TFC-2. To identify DEGs for GO and KEGG analyses,

we used the MAST framework as implemented in the Seurat Find-Markers() function and set the "latent.vars" argument to "nFeature_RNA," which is equivalent to adjusting for the cellular detection rate[53]. We selected genes with an average log-fold change >0.5 and adjusted *P*-values < $10^{-20}$. Gene Ontology (GO) analysis and pathway enrichment of DEGs (TFC-1 and TFC-2) were analyzed using Metascape (http://metascape.org) with default parameters and performed based on the KEGG pathway[54] and GO_Biological Processes. Mouse genes were converted to their human orthologs. Then GSEA (v4.0.3)[55] was performed between TFC-1 and TFC-2, and the hallmark gene sets were used as the database. The permutation type was set to phenotype, and the number of permutations was 1000. *P*-values of both over-representation analyses and the GSEA were adjusted to FDRs, and a gene set was considered significant if FDR < 0.05.

**Trajectory analysis.** TFCs clusters and accompanying nonnormalized gene expression count data from the final integrated Seurat object were used as inputs to create Monocle v2 newCellDataSet[31,56]. Differential expression between clusters was calculated using differentialGeneTest function in Monocle. DDRTree method was used for dimensionality reduction, and the pseudo-time trajectory plot was generated using Monocle's plot_cell_trajectory function, where cells were colored by pseudo-time placement. Differential expression analysis among different TFC types was performed, and the top 100 pseudo-time-dependent genes were displayed for visualization and GO analysis and pathway enrichment.

**Ligand–receptor interaction analysis.** The average expression level and *P*-value for each ligand–receptor pair were obtained using the statistical framework of CellPhoneDB[36]. Briefly, reference list of human ligand–receptor pairs previously published[37] were selected and converted into mouse orthologs using the Ensembl biomaRt package[57].The ligand–receptor score was defined as the mean of the average log-normalized expression of the receptor gene in one cell type and the average log-normalized expression of the ligand gene in a second cell type. Permutation test was used to identify enriched ligand–receptor interactions. For each of the 1000 permutations, we randomly shuffled the cell-type and time-point labels and calculated the interaction scores for all possible combinations. Repeating this 1000 times generated a null distribution of the interaction scores for each ligand–receptor pair. We compared the actual interaction scores of specific [ligand cell A – receptor cell B] to the null distribution scores and *P* values were calculated as the proportion of null scores.

**Spatial transcriptome analysis of mice thyroid tissues**

**Mouse thyroid tissue collection and slide preparation.** For ST transcriptional analysis, dissected thyroid cartilage area tissues from two mice were gently washed with cold phosphate-buffered saline (PBS) (Gibco, catalog no. 20012–043). Then the tissues were placed in an OCT-filled mold and snap-frozen in isopentane and liquid nitrogen. Tissue blocks were stored at −80 °C for a maximum of 1 month before use. The Visium Spatial Gene Expression Slide (10 × Genomics, PN-1000185, Pleasanton, CA, USA) includes four capture areas (6.5 mm × 6.5 mm), each capture area has ~5000 gene expression spots, each spot with primers that include Illumina TruSeq Read 1 (partial read 1 sequencing primer), 16-nucleotide (nt) spatial barcode (all primers in a specific spot share the same spatial barcode), 12-nt UMI; 30-nt poly(dT) sequence (captures polyadenylated mRNA for cDNA synthesis).

**RNA integrity number (RIN) detection.** RNeasy Mini Kit (QIAGEN, catalog no. 74104) was used to test the RIN. After taking 10 slices of 10 mm thick cryosections, RNA was extracted and analyzed by RNeasy Mini Kit immediately. An RIN of ≥7 is qualified.

**Optimization of the permeabilization time.** Before using a new tissue for generating Visium Spatial Gene Expression libraries, the permeabilization time was optimized. Briefly, the Visium Spatial Tissue Optimization workflow included placing thyroid tissue sections on seven-capture areas on a Visium Tissue Optimization slide (10 × Genomics, PN-1000186, Pleasanton, CA, USA). The sections were fixed, stained, and then permeabilized for different times (0, 3, 6, 12, 18, 24, and 30 min). Tissue permeabilization with 24 min results in maximum fluorescence signal in the thyroid region.

**Tissue cryosectioning, fixation, staining, and bright-field imaging for ST.** For cryosectioning, both the tissue block and the Visium slide were equilibrated inside the cryostat for 20 min at 16 °C before sectioning. Transverse sections through the entire bilateral thyroid were cut at a thickness of 10 μm and immediately placed on the Visium array slide. Then the section was fixed for 10 min with 3.6% formaldehyde in PBS, stained with hematoxylin and eosin, and mounted in 80% glycerol for bright-field imaging. Imaging was taken on a Leica SCN400 F whole-slide scanner at 40 × magnification.

**Construction of cDNA libraries and sequencing.** After bright-field imaging, the thyroid ST slide was permeabilized at 37 °C for 24 min and washed with 100 μl of 0.1 × SSC buffer. Then the on-slide reverse transcription (RT) reaction was performed at 53 °C for 2 h. Second-strand synthesis was subsequently performed on slide for 15 min at 65 °C. Following second strand synthesis, the thyroid sample was transferred to tubes containing tris-HCl (1 M; pH 7.0) for amplification and library construction. Library quality was assayed using a 1 μl sample on a Bioanalyzer High Sensitivity chip (Agilent, catalog no. 50674626). 10× Genomics Visium libraries were prepared and sequenced on Illumina NovaSeq 6000 (Illumina, San Diego, CA, USA) using a 28 bp + 120 bp paired-end sequencing mode. Sequencing data were processed using the Space Ranger pipeline v.1.0.0 (10 × Genomics). This generated data sets containing 1128 spots.

**ST data processing and cell-type estimation.** Raw fastq files were processed using the Space Ranger (V1.2.1), and Seurat (V3.2.2) was used to process the Space Ranger output files. For the gene-spot matrixes generated by Space Ranger, some routine statistical analyses were performed first, including calculating the number of the detected UMIs and genes in each spot. On this basis, the spots with extremely low UMIs or genes were removed, and mitochondrial and ribosomal genes were filtered. According to morphological staining, we excluded non-thyroid area spots. This QC generated data sets containing 454 spots.

After QC, we used the R package harmony (v1.0) to integrate the expression data of the two mice thyroid tissues and used the Seurat package (v3.1.5) to perform the basic downstream analysis and visualization. In detail, UMI counts in each spot were normalized by the total transcript count and then scaled by the median number transcript count across all spots. Then we identified 2000 highly variable genes according to their expression means and variances for PCA, which was performed to project the spots into a low-dimensional space (the first 20 principal components). Then, by setting the sample source as the batch factor and using the "Run-Harmony" function, we iteratively corrected the spots' low-dimensional PC representation to reduce the impact of individual differences. After this step, the corrected PC matrixes were used to perform unsupervised shared-nearest-neighbor (SNN)-based clustering and UMAP visualization analysis. In addition, to extract marker genes specific to each of all or selected clusters, a two-tailed Student's *t* test was used, *P* < 0.001. We identified TFC (*Tg*, *Tpo*), PTC (*Pth*, *Chga*), and adipocyte (*Fabp4*, *Ucp1*), which represent all major cell types in the thyroid slide.

**Determination of cell-type enrichment and depletion by multi-modal intersection analysis (MIA).** MIA was done as previously reported. Briefly, in the scRNA-seq data, we defined the gene sets for each cell type by identifying genes with statistically higher expression in the cells annotated to that cell type and compared their expression to that in the remaining cells ($P < 10^{-20}$, two-tailed Student's *t* test). For the ST data, we identified genes with significantly higher expression in each spatial region relative to the others ($P < 0.001$, two-tailed Student's *t* test). With the specific gene sets extracted from the scRNA-seq and ST datasets, the overlap between each pair of cell type-specific and tissue region-specific gene sets was analyzed using MIA. The results were displayed using a hypergeometric test, which is used to infer the significance of the intersection of genes specifically expressed in each cell type identified by scRNA sequencing and genes specifically expressed in each tissue region by ST sequencing. The significance of the overlap between ST genes and cell type marker genes was quired using hypergeometric cumulative distribution, with all genes as the background to compute the *P*-value. In parallel, we tested for cell-type depletion by computing $-\log_{10}(1 - P\text{-value})$. By this method, the "MIA map" was obtained by analysis of all pairs of cell type- and thyroid region-specific genes.

**GO and GSEA and analysis.** For GO analysis of TFC-central and TFC-peripheral cell types by using the Metascape tool (http://metascape.org)[58] with default parameters, we took all marker genes with an average log-fold change >0.5 and adjusted *P* values < 0.001. Functional enrichment analysis was performed based on the KEGG Pathway and Gene Ontology Biological Process. The statistical significance was tested by hypergeometric test and adjusted by Benjamini-Hochberg *P* value correction algorithm. After gene conversion from mouse to human, GSEA was done using the fgsea R package, using the MSigDB Hallmark gene sets. The permutation type was set to phenotype and the number of permutations was 1000. *P*-values of both over-representation analyses and the GSEA were adjusted to FDRs and a gene set was considered significant if FDR < 0.05.

**Experiments carried out in mouse models**
**Intraperitoneal injection of NF-κB inhibitors.** TPCA-1 (Selleck, 507475-17-4) with a concentration of 10 mg/kg was used for the mice's intraperitoneal injection.

**Determination of TSH and thyroid hormone levels.** Serum TSH, Thyroxine, and Triiodothyronine concentrations were measured using a commercial ELISA kit (CUSABIO: P12656, Cloud-Clone Corp: CEA452Ge, and CEA453Ge) according to the manufacturer's instructions.

**Polychromatic immunofluorescence staining with tyramide signal amplification fluorophore.** Polychromatic immunofluorescence staining was performed using a four-color multiple fluorescent immunohistochemical staining kit (Absin, abs50012) according to the manufacturer's instructions. Briefly, paraffin sections of the mice's thyroid tissues were dewaxed with xylene and rehydrated with ethanol. After antigen retrieval using sodium citrate, the slices were blocked and incubated with primary antibody for 1 h at room temperature. Incubation with horseradish peroxidase (HRP)-labeled secondary antibody was followed by tyramide signal amplification (TSA) fluorophore staining. For the next round of protein staining, the slices were boiled in antigen retrieval buffer to remove residual unbound antibodies from the previous round. After counterstaining with DAPI and mounting in glycerol and gelatin, the slices were imaged for 5 days. Antibodies used: E-cadherin (Servicebio, GB12083, 1:500); β-catenin (Sigma, C2206,

1:1000); NKX2-1(Servicebio, GB14157, 1:200); P65 (CST, 8242, 1:400); Phospho-NF-κB p65 (Ser536) (CST, 3033, 1:800); CD68 (Servicebio, GB113109, 1:100); TNF-α (Servicebio, GB11188, 1;200); Vimentin (Servicebio, GB11192, 1:200); MCAM (Invitrogen, 14-1469-82, 1:200); Phospho-IKKα/β (CST, 2697, 1:50); Thyroxine (Abcam, ab30833, 1:50) and IL-1 beta (Servicebio, GB11113, 1:800).

**RNA-scope combined with or without immunofluorescence analysis.** The RNA-scope assay was performed on thyroid sections using the RNA-scope Multiplex Fluorescent Reagent Kit v2 (323100; Advanced Cell Diagnostics (ACD), Hayward, CA, USA). Briefly, tissue sections were deparaffinized with xylene and 100% ethanol and incubated with hydrogen peroxidase for 10 min at room temperature. They were then treated with target retrieval reagent for 30 min at 98–102 °C and Protease Plus (Pretreatment kit 322381; ACD) for 15 min in a HybEZ hybridization oven (ACD) at 40 °C. The slides were hybridized with the probes for Ms-*pax8* (574431-C2), Ms-*tnfrsf1a* (438941), and Ms-*map4k4* (446661) in an oven at 40 °C for 2 h. After hybridization, the slides were subjected to signal amplification by incubation with AMP1, AMP2, AMP3, and C1-HRP (in that order) using a detection kit, and the hybridization signal was detected using Opal 570 (FP1488001KT; Akoya Biosciences, USA) (1:1500). If immunofluorescence analysis was performed, the slides were blocked with HRP-block buffer and then subjected to TSA fluorophore staining.

**Tissue immunohistochemical staining assays.** Immunohistochemical staining was performed as instructed. In brief, paraffin sections of 5 μm thickness were prepared. The sections were deparaffinized, treated with 3% $H_2O_2$ for 10 min, autoclaved in 10 mM citric sodium (pH 6.0) for 30 min for heat-induced antigen retrieval, and then incubated with primary antibodies at 4 °C overnight; this was followed by incubation with biotinylated secondary antibody for 1 h at room temperature. Finally, 3,3-diaminobenzidine tetrahydrochloride was used as a coloring reagent, and hematoxylin was used as a counterstain for the nuclei. Antibodies used includes: E-cadherin (Servicebio, GB12083); TPO (Servicebio, GB14160); P65 (CST, 8242, 1:400).

**In vitro thyroid folliculogenesis on Matrigel and immunofluorescence.** Mouse thyroid follicles were isolated according to a previously published protocol[59], with minor modifications. Thyroid lobes were dissected from 3-week-old mice. The lobes were collected in a 1.5-ml microcentrifuge tube containing 1 ml of digestion medium, which consisted of 100 μ/ml collagenase I, 100 μ/ml collagenase II, and 1 μ/ml dispase, dissolved in DMEM/F-12. Enzymatic digestion was carried out for 1 h in a 37 °C water bath, with manual shaking every 15 min. After digestion, Single-cell suspensions were obtained by filtering through a 40 μm cell-Strainer nylon mesh (BD). The cells were seeded on Matrigel (Corning, 356234) and cultured in F12 medium supplemented with 5% FBS, 5 μg/mL transferrin, 10 μg/mL bovine insulin, 360 pg/mL hydrocortisone, 10 ng/mL somatostatin, 100 ng/mL Gly-His-Lys, 100 μg/mL penicillin-streptomycin, 2.5 mg/mL NaHCO₃, and 1 mμ/L TSH. To visualize the follicles in 3D Matrigel, the cells were fixed for 15 min in 4% paraformaldehyde (PFA), washed with quenching solution (100 mM Glycine in PBS), and permeabilized with permeabilization buffer (0.5% Triton X-100 in PBS) for 3 h at room temperature. Then after washing and blocking, primary and secondary antibodies were successively incubated for 1–2 days at 4 °C.

**Experiments carried out in zebrafish models**
**Zebrafish husbandry.** Zebrafish maintenance and staging were performed using standard protocols[60]. This study was approved, and the methods were carried out in accordance with the approved guidelines. Transgenic zebrafish lines *mpeg1:eGFP*, *mpo:eGFP*[61], *flk1:*

*eGFP*[62] and *tg:eGFP* were obtained from China zebrafish resource center. *Tg:mCherry* was created by our lab[63]. The *cebpa*^ΔCA zebrafish lines was a gift from Jun Zhu's lab[64]. The plasmid used for creating Tg(*nfkb*: *eGFP*) was a gift from John Rawls (Addgene plasmid # 44922)[33]. For Tg(*tnfa: dendra*) transgenic lines, the TNFa promoter (Gene ID: 405785) was amplified from zebrafish genomic DNA[39] and then cloned into Tol2 vector backbone (Addgene, plasmid # 51462) upstream of dendra2. The plasmids were linearized and co-injected with transposon mRNA at the zebrafish one-cell stage. Stable transgenic lines were finally obtained by offspring fluorescence screening.

**Chemical treatment on zebrafish embryos.** Zebrafish embryos were incubated in an E3 medium containing TPAC1 (Selleck, 40 μM), BMS-345541 (Selleck, 20 μM), lenalidomide (Selleck; S1029; 10 μM), and NSC 23766 trihydrochloride (Abcam, 10 μM) from 24hpf until collection.

**Constructs.** TRAF2 was generated from human cDNA of open reading frame sources using PCR. Human NFKBIA (IκBα) with S32/36 A mutations[35] was cloned under the control of the *Tg* promoter. For transiently overexpression IκBα in the thyroid cells of zebrafish embryos, wild-type embryos were co-injected with 30 ng/μL of Tol2 transposase mRNA with either 30 ng/μL of Tol2 *tg*:nfkbia or Tol2 *tg*:mCherry at the one-cell stage. Capped mRNA encoding for Tol2 transposase was generated by in vitro transcription using mMessage mMachine SP6 kit (Ambion), and NotI linearized pCS-zT2TP plasmid[65] as a template.

**Whole-mount in situ hybridization.** Briefly, after fixation, the zebrafish embryos were first treated with 8% H2O2 and 50% formamide/ 2XSSCT (9:1) for 20 min to avoid pigmentation. After which the samples were dehydrated using methanol and rehydrated on the day doing WISH. Samples were permeabilized and hybridized using Digoxin labeled RNA probes as indicated. Antisense probes for *tg* and *tpo* were transcribed in vitro using the T7 transcription kit (Roche) from a plasmid containing zebrafish thyroglobulin or thyroid peroxidase. After alkaline phosphatase conjugated anti-Digoxigenin antibody incubation overnight, the signals were detected using NBT/BCIP substrate. Images were captured using the Nikon SMZ25 microscope.

**Zebrafish whole-mount immunofluorescence assay.** In brief, embryos were fixed in 4% PFA overnight at 4 °C and were then dehydrated with methanol at −20 °C. The dehydrated embryos were washed three times with PBST, followed by proteinase K treatment. The embryos were re-fixed with 4% PFA at room temperature for 20 min and washed three times with PBST. After blocking for 1 h at room temperature with blocking solution (2 mg/mL bovine serum albumin, 10% FBS, 0.3% Triton-X100, and 1% dimethyl sulfoxide in PBST), antibodies were added to the blocking solution and incubated at 4 °C overnight. The samples were then washed thrice with PBST and incubated with the secondary antibody for 2 h. Finally, after washing thrice with PBST, images of the embryos were captured using confocal microscopy. The antibodies used were anti-thyroxine (Abcam, ab30833; 1:50) and anti-TNF-α (Santa Cruz, sc-133192; 1:100).

**Confocal microscopy.** Live embryos were anesthetized and mounted on dishes with 1% low-melting agarose. Confocal images were captured using a Nikon A1 confocal laser microscope. The images were analyzed using Nikon confocal software and Image J.

**Real-time qPCR.** After treatment with DNase I (Ambion), RNA was prepared using the TRIzol reagent (Invitrogen, H10522) and then subjected to cDNA synthesis using a cDNA synthesis kit (ABI). Real-

time PCR was performed using a Fast Start Universal SYBR® Green Master Rox probe (Roche Applied Science, 13800300) and Mastercycler thermal cycler (Eppendorf, 22331).

**Experiments carried out in vitro cell lines**
**Cell culture and reagents.** Cells were cultured in a humidified incubator at 37 °C in the presence of 5% $CO_2$. Nthy-Ori3 (Sigma, 90011609) and HL60 cells (Sigma, 98070106) were cultured in RPMI-1640 medium supplemented with 10% FBS, 100 IU/mL penicillin, and 100 µg/mL streptomycin (Gibco). TNF-α (Sigma; T0157) and lenalidomide (Selleck; S1029) were used for cell treatments.

**Western Blot.** Proteins from cell lines in lysis buffer (Beyotime, P0013) and protease inhibitor cocktail (Roche, 04693132001) were stored on ice for 40 min as instructed. The antibodies used for western blot were as follows: mouse anti-IQGAP1(Invitrogen, 33-8900; 1:1000), rabbit anti-Vimentin (Servicebio, GB11192, 1:1000), rabbit anti-MMP1 (Thermo, PA5-115581, 1:1000), rabbit anti-MMP2 (Servicebio, GB11130, 1:500), mouse anti-GAPDH (Sigma, WH0002597M1, 1:2000).

**Statistical analysis**
As with non–scRNA-seq studies, data were presented as the mean ± standard deviation (SD). Group comparisons of normally distributed data were performed using an unpaired Student's $t$ test. One-way ANOVA, followed by Tukey's test was used for multiple groups comparison. Fisher's exact test was used for testing the independence of categorical data. Statistical analysis was performed in GraphPad Prism 7.04 (GraphPad Inc.). Statistical significance was set at $P < 0.05$.

**Reporting summary**
Further information on research design is available in the Nature Portfolio Reporting Summary linked to this article.

## Data availability
All data associated with this study are presented in the paper and the Supplementary Materials. This study did not generate any unique code. All software and algorithms used in this study are freely or commercially available and are listed in the Methods section. The sc-RNA seq data and the spatial seq data generated in this study are available in GEO database with accession code GSE231954. The resulting fastq files were aligned to the mouse reference genome (mm10) (https://www.ncbi.nlm.nih.gov/datasets/genome/GCF_000001635.20/). Source data are provided with this paper.

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

## Acknowledgements

We are grateful for Dr. Hao Yuan and Pro. Jun Zhu for providing *cebpa*$^{\Delta CA}$ zebrafish lines. This work was supported by grants from National Science Foundation of China (No. 82270826 H.-D.S., 82170802 R.-M.Y., 81661168016 H.-D.S., 81770786 H.-D.S., 81803014 R.-M.Y., 81870540 J.L., 81870537 S.-X.Z.), National Key R&D Program of China (2017YFC1001801 H.-D.S.), China Postdoctoral Science Foundation (No. 2019M651518 R.-M.Y.), Innovative research team of high-level local universities in Shanghai (SHSMU-ZDCX20212501 S.-X.Z.), Two-hundred Talent (No. 20161318 S.-X.Z.).

## Author contributions

H.-D.S. developed the concept of this study. R.-M.Y., M.Z., F.Y.W., P.-H.T., J.-X.Z., F.C., L.Y., J.L., M.-M.G. and G.-Q.G. performed most of the experiments in mice, zebrafish and cell lines. The 10 × single-cell and ST sequencing data were analyzed by Y.-T.S., M.Z., R.-F.Y., Z.W. and S.-Y.S. H.-D.S., Y.Z., M.Z., S.-X.Z. and R.-M.Y. prepared the draft and final version of the manuscript. All authors reviewed the results and approved the manuscript.

## Competing interests

The authors declare no competing interests.
