## [Peer Review File · Nature Communications]

REVIEWER COMMENTS

Reviewer #1 (Remarks to the Author):

The Authors produce evidence in animal models (zebrafish and mice) of the interaction of myeloid cells and TFC during thyroid morphogenesis. They also found biallelic or hemizygous variations in genes involved in TNF- α NF-kB pathway in several CH patients.

The data are original and provide unprecedented insights into the mechanisms providing thyroid follicle growth. The results are obtained with up-to-date technologies and their illustrations are clear.

Comments:

- The Authors should try to give an explanation for the thyroid defect found in models upon their pharmacological/genetic manipulations: is this the result of a reduced proliferation or increased apoptosis? Are the same events occurring in *cebpa*^{-/-} line and in pharmacological manipulations?
- Could dysfunctional immune cells (eg, macrophages) involved in a sort of TFC destruction? This latter mechanism would be relevant to be investigated in view of the association with immunological disorders in patients.
- Fig 2d-j show reduced motility and mesenchimal markers (Vim, Mcam or map4k4) in mice. Variations in TFC-1 vs TFC-2 cell populations should be shown.
- Some vascular defects appear to occur in Supplemental fig 12, raising some doubts on a thyroid cell specific effect. The Authors should give more insights on the conditions of heart, great vessels in their models.
- How can the Authors explain the large predominance of variants in *DUOX2* defects in their CH population (>2/3 in figure S13)?
- Positive autoantibodies are almost always negative in the CH population (either in TD or GIS patients). How can the authors reconcile their findings, pointing to a general immunological dysfunction associated with TNF- α NF-kB pathway defects, with such a relevant fact?
- The severity of CH (thyroid function tests) in the patients with NF-kB pathway defects must be shown.

Other comments:

- English revision is required (some sentences are too long, and there is a general misuse of adverbs and adjectives)
- Introduction: "...the molecular mechanisms underlying TD are unknown". This sentence is incorrect and should be revised because it does not acknowledge the work of several researchers in the field.

- Sentences at lines 253-255 and 263-265 should be revised

Reviewer #2 (Remarks to the Author):

Although the effect of NF κ B on thyroid development and differentiation has been reported previously (DOI: 10.1089/thy.2020.0208; Endocrinology 139: 1715–1722, 1998), in this manuscript, they demonstrated that TNF α , which leads to the increase of NF κ B, came from myeloid cells. A reasonable explanation for the source of NF κ B in thyroid cells was also given. Several different thyrocytes were identified, and the typing of TFC1 and TFC2 is indeed very meaningful. TFC1 may be early thyroid epithelial cells or incompletely differentiated thyroid epithelial cells, which tend to be in developing; TFC2 should be mature thyroid cells that tend to be functional. At last they verify the mutations of NFB pathway related genes in part of CH patients.

This is a very significant study, which has discovered a new pathogenesis pathway of congenital hypothyroidism and its possible mechanism.

Suggestions and comments:

1. Thyroid transcription factor 1, TTF-1, is an irregular name. The formal name should be NK2 homeobox 1, NKX2-1. TTF1 is the formal name of transcription termination factor 1
2. In Figure 1I, the label of the x-axis is incorrect. It should be E-catenin instead of β -catenin.
3. In the early stage, as the thyroid grows, the cell density decreases, mainly due to the formation of follicles. Under the same field of vision, the number of thyroid cells in the 30-day mice is necessarily less than that in the 5-day mice, so whether there is a problem in the process of calculating the cell ratio
4. Figure 3G, TNF- α -NF- κ B signaling pathway related genes are relatively highly expressed in TFC central, although some p values are not less than 0.05. I wonder if the authors didn't include genes with relatively low expression, or is that the result?
5. How are central and peripheral thyroid cells identified in spatial transcriptome studies? I observed that some peripheral cells in the left lobe appeared to be incorporated into the central thyroid cells. Furthermore, the thyroid does not have distinct tissue types (such as cortex and medulla) as the kidney and spleen do. Also, it isn't as hierarchical as the skin or the central nervous system, so does a central and peripheral division to be reasonable?
6. Considering that there are many kinds of cells in vivo, in order to verify the effect of myeloid cells on thyroid cells, I suggest to do co-culture experiments in vitro?
7. For TNF- α -NF- κ B signaling pathway related gene mutations study in population, it is suggested to score the deleteriousness of these gene mutations (according to mutation frequency, conservatism, mutation site location, mutation type, and possible impact on protein structure).

Reviewer #3 (Remarks to the Author):

Remarks to the authors

In this study, the authors identified a subtype of thyrocytes (i.e., TFC-1) that was located in the center of thyroid tissues of postnatal mice using 10X single-cell RNA-seq and spatial transcriptome sequencing technologies. This subtype of thyrocytes exhibited high migratory capacity by TNF- α -induced NF- κ B activation, which may drive thyroid folliculogenesis. The authors further demonstrated that TNF- α -induced NF- κ B activation of thyroid epithelial cells was triggered by myeloid cells in thyroid tissues, as shown by the cell-cell interaction analysis of single-cell data and experimental validation. In addition, they found that the dysregulation of the TNF- α -NF- κ B pathway was associated with the pathogenesis of thyroid dysgenesis (TD).

These datasets and findings will be an important resource for the field. However, it is a sub-optimally written paper and the omics data analysis is not comprehensive enough. Specific comments and concerns are as follows:

Major concerns:

1. Since the study is designed to investigate the mechanisms underlined transformation of thyroid primordium into mature follicles by dynamical observation of the process of thyroid follicle formation, the authors obtained single-cell datasets from the thyroid tissues of mice on postnatal 5 days, 10 days, 20 days, and 30 days. However, the developmental trajectory inference in single-cell data analyses is completely ignored. At least the authors should perform pseudotime analysis on thyroid follicular cells to assess the alterations of the transcriptome, the potential regulators, and the signaling pathways in thyroid follicular cell subsets during different postnatal days.

2. It is understandable that TFCs were selected for in-depth analysis. However, the information on the subpopulation distributions of a large number of cell types (~25000/30000) and their compositional changes at different postnatal timepoints was ignored. The authors should provide a paragraph in the main text with an overall description of the single-cell transcriptome profiling of thyroid tissue cells.

3. The description of findings in the analyses of scRNA-seq data is incoherent.

–Line 95-98. What is the point of this result supposed to infer? How does it relate to the above context, especially after adding the timepoint information?

–Line 98-102. Based on the results that TNF- α -NF- κ B was significantly enriched in TFC-1 compared to the TFC-2 subtype, the authors could not obtain an inference of time-dependent activation of the TNF- α -NF- κ B signaling pathway.

I am not sure how many similar incoherent issues exist elsewhere.

4. Related to the 1st comment, the authors obtained single-cell datasets from the thyroid tissues of mice on postnatal 5 days, 10 days, 20 days, and 30 days according to the study design. Why did the authors perform 10X spatial transcriptome sequencing on thyroid glands at postnatal 10 days only? It seems to me that the authors had no interest in investigating transcriptomic alterations and spatiotemporal dynamics in the thyroid tissues of distinct postnatal time points.

5. Did the authors take the spatial context into account in their data analysis of cellular interactions? Several integration methods (e.g., Tangram (Biancalani et al. Nat. Methods 2021), Cell2location (Kleshchevnikov et al. Nat. Biotech. 2021), RCTD (Cable et al. Nat. Biotech. 2021)) can leverage cell type profiles learned from single-cell RNA-seq to decompose cell type mixtures in spatial transcriptomics, and estimate the spatial co-localization between distinct cell types, which can help facilitate cell-to-cell interaction analysis with spatial information.

Minor concerns:

1. I strongly recommend that the overall language of the main text should be polished with the assistance of professional English editors to improve the cogency and organization of the manuscript.

2. What doublet detection and removal method was used for the analyses of scRNA-seq data? This is not stated.

3. It will be useful to show the UMAP visualizations or clustering consistency and findings following batch correction using one more method (e.g., Harmony(Korsunsky et al., Nat. Methods 2019), LIGER(Welch et al. Cell 2019)) for the analyses of scRNA-seq data. Batch effects if not adequately addressed will affect findings and conclusions.

4. It is not appropriate to show different gene combinations in different figure panels (Fig. 2m, Fig. S3e, Fig. 3g) for the same TNF- α -NF- κ B signaling pathway.

5. What did the results of the PCA indicate? Fibroblast and epithelial cells are not distinguished from each other in Fig. S2i.

6. Line 87-90, Why are the changes in the expression of several transcription factors mentioned in the paper within the authors' expectations? At least, references should be added.

7. Line 112, "Although the low resolution of ST-seq that one spot containing 50 cells made it difficult to distinguish single cells that were scattered in the thyroid tissue", did the authors perform an analysis to assess the average cell number per spot in ST slice for thyroid gland?

—When our group analyzed the average cell number per spot in our research system (different tissues from thyroid glands), one spot contained ~10-20 cells (10X Visium).

8. In Fig. 3c, I noticed that there are only 13 upregulated DEGs in TFC-2 cells compared with TFC-1 cells, while 2556 upregulated DEGs in TFC-1 cells compared with TFC-2 cells in scRNA-seq data. Please double-check the analyzed data and provide a supplementary table for the DEGs between TFC-1 cells and TFC-2 cells.

9. The panel number of Fig. 3 is not provided. There is a similar issue in Fig. 6.

Reviewer #4 (Remarks to the Author):

The authors analysed an impressive amount of data to describe a link of the TNF- α -NF- κ B pathway in development of the thyroid gland. With the developmental process being largely undescribed this manuscript provides important information. Unfortunately, the data is overwhelming and the presentation does not always help to digest the information. It would help if the authors reduce data in the main article and focus on the main points they want to convey with a better structure of the text.

A concern is that the authors might be looking at two different processes: migration of the whole gland to the correct anatomical position (eg in some Zebrafish data, <72hpf) and migration of individual follicles (in new born mice and older zebrafish) within the thyroid gland? While regulation might be involving similar pathways, it would make sense to pull these two processes apart?

The link with human recurrent biallelic variations of core genes in TNF- α -NF- κ B pathway is rather weak analysing tables "Table S9-S10":

Prediction shows IKBKB and IKBKG variations are predicted benign, two carriers have no gland phenotype, two other either Thyroid dysgenesis or Autoinflammation.

CHT00217, IRAK1: with Thyroid dysgenesis , but no autoimmune, is predicted benign

CHT00189,RBCK1: prediction not uniform, Autoinflammation, with no thyroid phenotype

Most references (35 to 40) cited in the discussion refer to autoimmune hypothyroidism, what can be considered a different disease mechanism, while they might influence the progress of the phenotype, they are not reported to be linked with migration/development described here.

Investigation of large data sets of whole gene sequencing with a description of the thyroid phenotype/autoimmune disease might be helpful.

In the discussion the authors mention that here are potential alternative described pathways (eg in some references in introduction) that could be involved, this can easy be investigated using eg RNA-seq data, other examples

NIS can be negatively regulated by TNFalpha <https://pubmed.ncbi.nlm.nih.gov/32049985/>

Was NIS expression assessed in their model?

In Ref 10 SOX9 shows, in an earlier stage of development, a similar expression pattern.

Is there an acute (1-5 days?) effect on thyroid function with drug treatments used, this before thyroid follicle development is affected?

Absence of figure numbers on main text figure and changing colours of the targets eg fig 5b-c-d CD68 change, also in other figs, makes it harder to read. Not all data has been quantified, this would be sensible to come to a conclusion.

P4

Line56-58: “found folliculogenesis is not synchronous because peripheral thyrocytes formed a monolayer earlier than the central thyrocytes, which initially appeared as closed spheres (Fig. 1b).”

This is not obvious from this picture, The TPO staining in Supplementary Fig. 1b are much more convincing, while the E-cadherin staining with the same days of maturation, does not show the same pattern. Quantification can clarify this.

Fig 4A suppl fig 6:

According to your ref 6 migration of TP is not finished until 72hpf, so 2.5 days and the 52-62.5hpf window fall well within that. GFP and mCherry are stable proteins (day-hours?) and might not be reflecting the correct response time of gene expression?

P7

line 160 “After treatment with TPCA1, thyroid function was compromised”

What is the thyroid hormone status in the zebrafish? Is there an acute (1-5 days?) effect on thyroid function after different drug treatment?

TPCA-1 is a direct dual inhibitor of STAT3 and NF- κ B, did the authors look at the effect on STAT3 related pathways?

P8 line 172-176

“We further specifically overexpressed the constitutively active form of NFKBIA (I κ B α) in thyrocytes under the thyroglobulin promoter, which retains NF- κ B members in the cytosol to resist NF- κ B activation. Transient overexpression of this plasmid in thyrocytes induced similar thyroid abnormalities, confirming the intrinsic role of NF- κ B signaling in thyroid development (Supplementary Fig. 8e,f).”

Did the authors show in this model similar outgrowth. No details about transfection, methods – efficiency - ...

explain contradiction in Supplementary Fig. 8e,

“Representative images showing thyroid specific inhibition of NF- κ B activation by constitutively overexpression nfkbia under thyroglobulin promoter”

“Transient overexpression of this plasmid” in text versus “constitutively overexpression” in Supplementary Fig. 8e,

P8

Line 174-5 “Transient overexpression of this plasmid in thyrocytes induced similar thyroid abnormalities”

I did not find the Transient overexpressing method in zebrafish.

P8

“184 Myeloid cells in the developing thyroid tissue secrete TNF- α to trigger the NF- κ B
185 activation of TFCs”

Refer to species used in the beginning of this section

Supplementary Fig. 9: Refer to species used

P9, line 206-213

The Authors need to take care comparing events in zebrafish at 2dpf and 5dpf. At 2dpf the TP is still migrating to its final position in the zebrafish, according to ref 6. At day 5dpf it is more likely a maturation process, similar to what is seen in mice.

fig 5b-c-d CD68 does not surround p65 positive cells but might be in close proximity, fig 5c, suggest the other way round. Quantification?

P10-11,

“248 Recurrent biallelic variations of core genes in TNF- α -NF- κ B pathway identified
249 from CH patients with an unexplained etiology”

Can we assume that genes from Fig2 - fig 3 not appearing in table S7 did not show relevant variations?

If I understand correctly only 4 individuals from 168 with TNF- α -NF- κ B pathway link have also an autoimmune disease? What is the incidence of autoimmune disease in the 113/281 patients with no TNF- α -NF- κ B pathway link?

None of these genes (except NFKB1) appear in the TNF- α -NF- κ B regulated genes in fig 2m, fig3g, how are they affected?

REVIEWER COMMENTS

Reviewer #1 (Remarks to the Author):

The Authors produce evidence in animal models (zebrafish and mice) of the interaction of myeloid cells and TFC during thyroid morphogenesis. They also found biallelic or hemizygous variations in genes involved in TNF- α NF- κ B pathway in several CH patients.

The data are original and provide unprecedented insights into the mechanisms providing thyroid follicle growth. The results are obtained with up-to-date technologies and their illustrations are clear.

Comments:

- The Authors should try to give an explanation for the thyroid defect found in models upon their pharmacological/genetic manipulations: is this the result of a reduced proliferation or increased apoptosis? Are the same events occurring in *cebpa*^{-/-} line and in pharmacological manipulations?

Response: That is an important issue. We checked the proliferation and apoptosis status in zebrafish embryos. As with both TPCA-1 treated and *cebpa*^{-/-} line, the number of apoptotic thyroid cells per thyroid gland were slightly increased compared with corresponding controls (see revised Fig S9c and S12a). Otherwise, the numbers of proliferating thyroid cells were relatively unchanged.

- Could dysfunctional immune cells (eg, macrophages) involved in a sort of TFC destruction? This latter mechanism would be relevant to be investigated in view of the association with immunological disorders in patients.

Response: That is really an awesome question and is also what we want to figure out. In the present study, we try to discriminate the reason behind thyroid dysfunction (intrinsic or extrinsic, as you said, destruction by immune cells) by specifically overexpression NFKBIA, which suppressed NF- κ B activation, in thyroid cells. Inhibition of NF- κ B activation in thyroid cells induced follicle formation defect in zebrafish, suggesting NF- κ B activation is intrinsically required for folliculogenesis (see revised Fig. S9e). However, for now, we could not exclude the possibility that dysfunctional immune cells also played a role in the CH patients carrying mutations in the genes of NF- κ B pathway. Combining the suggestions of both reviews and editors, we deleted the data regarding the CH patients in this revised manuscript. Our future detailed examination of the phenotypes regarding thyroid and immune dysfunction in patients and separately deleting NF- κ B signaling in immune and thyroid cells using a mice model could tangle this issue.

- Fig 2d-j show reduced motility and mesenchimal markers (Vim, Mcam or map4k4) in mice. Variations in TFC-1 vs TFC-2 cell populations should be shown.

Response: Thanks for your good suggestion. Gene expression levels of these motility related genes (Vim, Mcam, and Map4k4) were compared between TFC-1 and TFC-2 cell population (see revised Fig. S3e). As you can see, these genes were obviously higher expressed in TFC-1 as compared with TFC-2.

- Some vascular defects appear to occur in Supplemental fig 12, raising some doubts on a thyroid cell specific effect. The Authors should give more insights on the conditions of heart, great vessels in their models.

Response: This question is important as the pharyngeal vessels remodeling and late thyroid morphogenesis are not only spatiotemporally correlated, but also functionally linked. We examined the heart and great vessels surrounding the thyroid gland with either TPCA-1 or BMS treatment. The major pharyngeal vessels surrounding the thyroid tissue were labeled as the paper published by Robert Opitz et al. (“Transgenic zebrafish illuminate the dynamics of thyroid morphogenesis and its relationship to cardiovascular development”, PMID: 23022354). Although we could not exclude there might be slight changes, the heart and major vessel types surrounding the TP were all normally developed in zebrafish embryos treated with either TPCA-1 or BMS. (see revised Fig. S13c).

- How can the Authors explain the large predominance of variants in DUOX2 defects in their CH population (>2/3 in figure S13)?

Response: We are sorry for our unclear description. Our CH samples were divided into two groups according to whether or not carrying reported CH causative genes. In Fig. S13 of the original manuscript, we only included CH patients carrying the 21 reported CH candidate genes (defined etiology). Out of these patients, DUOX2 mutations could explain 2/3 of the incidence. However, if combing all CH patients (including those with undefined etiologies), only 1/3 of CH patients carry DUOX2 mutations.

- Positive autoantibodies are almost always negative in the CH population (either in TD or GIS patients). How can the authors reconcile their findings, pointing to a general immunological dysfunction associated with TNF- α NF- κ B pathway defects, with such a relevant fact?

Response: Yes, indeed, the positive autoantibodies in these CH patients is an interesting finding. It suggested that the CH patients with TNF- α NF- κ B pathway defects might be a novel subtype of CH that carried both thyroid and immune dysfunctions. However, to prove this, we need a more detailed examination of the immune cell functions in these

patients. Considering this is a relatively separate topic and also combining the suggestions of reviewers and editors, we removed the data regarding patients from the revised manuscript. We hope our future studies will answer your question.

- The severity of CH (thyroid function tests) in the patients with NF-kB pathway defects must be shown.

Response: As we deleted the results related to patients in the revised manuscript, we attached the thyroid function test in this rebuttal below.

Patient ID	TT3	FT3	TT4	FT4	TSH
CHT00091	1.31	3.68	4.20	0.84	>150.00
CHT00094*	NA	3.32	NA	0.69	1.15
CHT00160	NA	2.27	NA	0.43	>150.00
CHT00189*	NA	5.46	NA	1.82	>150.00
CHT00205	1.30	2.15	0.9	0.19	>150.00
CHT00217	1.33	3.80	3.90	0.71	94.40
CHT00240	0.60	1.37	0.88	0.10	>150.00
CHT00249	1.62	3.95	9.67	1.63	9.52
CHT00334*	2.38	5.44	10.50	1.37	11.17
CHT00376	1.63	3.87	0.47	0.08	12.95
CHT00422	NA	1.34	NA	0.41	>150.00
CHT00478*	NA	3.58	NA	1.09	5.83
CHT00541	1.47	3.88	8.40	1.25	12.67

*Examined after treatment. TT3: total serum T3 (normal range: 0.87-1.80 ng/ml); FT3: free serum T3 (normal range: 2.50-3.90 pg/ml); TT4: total T4 (normal range: 6.09-12.23 ug/dl); FT4: free T4 (normal range: 0.58-1.64 ng/dl); TSH, thyroid stimulating hormone (normal range: 0.34-5.60 uIU/ml); NA, not applicable.

Other comments:

- English revision is required (some sentences are too long, and there is a general misuse of adverbs and adjectives)

Response: Thank you for your suggestion. We have revised the manuscript regarding scientific writing and presentation, and a native English speaker has corrected the spelling, typing, and grammatical errors.

- Introduction: "...the molecular mechanisms underlying TD are unknown". This sentence is incorrect and should be revised because it does not acknowledge the work of several researchers in the field.

Response: Thanks for your scrutiny. We have changed this sentence to "Most CH cases

are caused by thyroid dysgenesis (TD), which results from defects in thyroid development. However, the molecular mechanisms underlying TD, especially in the maturation of thyroid follicles, remain largely unknown” in our revised manuscript.

- Sentences at lines 253-255 and 263-265 should be revised

Response: As we removed the data regarding the patients in this revised manuscript, these descriptions were also deleted.

Reviewer #2 (Remarks to the Author):

Although the effect of NFκB on thyroid development and differentiation has been reported previously (DOI: 10.1089/thy.2020.0208; Endocrinology 139: 1715–1722, 1998), in this manuscript, they demonstrated that TNFα, which leads to the increase of NFκB, came from myeloid cells. A reasonable explanation for the source of NFκB, in thyroid cells was also given. Several different thyrocytes were identified, and the typing of TFC1 and TFC2 is indeed very meaningful. TFC1 may be early thyroid epithelial cells or incompletely differentiated thyroid epithelial cells, which tend to be in developing; TFC2 should be mature thyroid cells that tend to be functional. At last they verify the mutations of NFB pathway related genes in part of CH patients.

This is a very significant study, which has discovered a new pathogenesis pathway of congenital hypothyroidism and its possible mechanism.

Suggestions and comments:

1. Thyroid transcription factor 1, TTF-1, is an irregular name. The formal name should be NK2 homeobox 1, NKX2-1. TTF1 is the formal name of transcription termination factor 1

Response: Thanks for pointing out this issue. We changed all TTF-1 of this study to NKX2-1 in our revised manuscript.

2. In Figure 1I, the label of the x-axis is incorrect. It should be E-catenin instead of β-catenin.

Response: Thank you for your careful correction. We have amended this mistake in our revised manuscript.

3. In the early stage, as the thyroid grows, the cell density decreases, mainly due to the formation of follicles. Under the same field of vision, the number of thyroid cells in the 30-day mice is necessarily less than that in the 5-day mice, so whether there is a problem in the process of calculating the cell ratio.

Response: Thank you for your good question. Yes, as you mentioned, with the increased formation of follicles and enlarged lumen size, the cell number under the same microscopic field of vision gradually decreases with age. So, we instead calculated the number of E-cadherin^{low} or β-catenin^{low} cells in every 100 Nkx2-1⁺ cells.

4. Figure 3G, TNF- α- NF- κ B signaling pathway related genes are relatively highly expressed in TFC central, although some p values are not less than 0.05. I wonder if the authors didn't include genes with relatively low expression, or is that the result?

Response: We are sorry for not describing these details clearly. Fig. 3g contains all TNF- α -NF- κ B signaling pathway-related genes (n = 117) that can be detected in TFC central and TFC peripheral by the 10 \times spatial sequencing. The results showed that most TNF- α -NF- κ B signaling pathway-related genes (n = 61) were significantly up-regulated in TFC central compared to TFC peripheral. We also observed that 12 genes (*Id2*, *Il6st*, *Sat1*, *Cdkn1a*, *Ifih1*, *Tnip2*, *Sod2*, *B4galt1*, *Trip10*, *Kdm6b*, *Spsb1*, and *Cebpb*) were down-regulated in TFC central as compared with TFC peripheral, but none reached significant level (All $P > 0.05$). In addition, we modified Fig. 3g to make the results clearer in our revised manuscript.

The revised Fig. 3g.

The dotted horizontal line indicates FC = 1, and the dotted vertical line show $P = 0.05$.

5. How are central and peripheral thyroid cells identified in spatial transcriptome studies? I observed that some peripheral cells in the left lobe appeared to be incorporated into the central thyroid cells. Furthermore, the thyroid does not have distinct tissue types (such as cortex and medulla) as the kidney and spleen do. Also, it isn't as hierarchical as the skin or the central nervous system, so does a central and peripheral division to be reasonable?

Response: That is an important issue regarding how central and peripheral thyroid cells were identified. In this study, the classification of TFC central and TFC peripheral cells was not pre-defined by the H&E staining of mice tissues. The thyroid spots were clustered into two groups by unsupervised learning and data clustering algorithm based on the gene expression profiles from the spatial transcriptome. When reflect the two clusters on the slice of H&E staining, we can see that the spots of one group is mainly located in the peripheral region of the thyroid gland, which we named TFC peripheral, and the other group is mainly located in the central region, which we termed TFC central. As a result, there are a few spots clustered in a way that is not consistent with their actual location in the thyroid tissue. However, they are generally well correlated.

As the Reviewer suggested, the thyroid neither has the same distinct tissue types (such as cortex and medulla) as the kidney and spleen nor the same hierarchy as the skin or the central nervous system. In fact, these descriptions are used for illustrating the maturation sequence of thyroid epithelial cell during thyroid development.

6. Considering that there are many kinds of cells in vivo, in order to verify the effect of myeloid cells on thyroid cells, I suggest to do co-culture experiments in vitro?

Response: As you suggested, we have examined this effect by co-culturing myeloid cell line HL60 and thyroid epithelial cell line Nthy-Ori3. The expression of the migration-related protein (IQGAP1, Vimentin, MMP1, and MMP2) were examined by western blot. The results show that in vitro co-culture with the neutrophil cell line HL60 enhanced the migration capacity of thyroid follicular cells, and these effects were weakened by lenalidomide (TNF- α inhibitor) treatment (revised Fig. S13d).

7. For TNF- α -NF- κ B signaling pathway related gene mutations study in population, It is suggested to score the deleteriousness of these gene mutations (according to mutation frequency, conservatism, mutation site location, mutation type, and possible impact on protein structure).

Response: Thank you for your good advice. Predicting the effect of these mutations by software would give us good hints for further detailed investigations of these mutations. However, combining the suggestions of all reviewers and editors, we removed the patients' data in this manuscript. We simply put this result below.

Patient ID	Gene	AA changes	SIFT_pred	Polyphen2		ExAC_EAS	esp6500_siv2_all	gnomAD_genome_EAS
				_HDIV_pred	_HVAR_pred			
CHT00249	ATM	Asn1650Ser	T	B	B	0.007	0.0006	0.0093
CHT00249	ATM	Thr2228Ile	D	P	P	NA	NA	NA
CHT00422	CARD10	Pro902Leu	T	B	B	0.003	3.2E-05	0.0006
CHT00422	CARD10	Arg307Trp	D	D	P	0	0.0001	0
CHT00205	CARD6	Ile194Val	D	B	B	0	0.0084	0
CHT00205	CARD6	Ser671Phe	D	P	B	0.002	0.0003	0.0055
CHT00240	IKBKG	Pro18Leu	NA	B	B	0	9.3E-05	0
CHT00217	IRAK1	Gly589Ser	T	B	B	0.002	9.2E-05	0.0019
CHT00334	IRAK1	Gly159Asp	T	P	B	0.001	4.6E-05	0.001
CHT00478	IRAK1	Ala470Thr	D	D	D	0.001	NA	NA
CHT00541	IRAK1	Val556Met	D	D	D	0.004	0.0001	0.0028
CHT00091	IKBKB	Ala708Thr	T	B	B	0.006	0.0004	0.0068
CHT00189	RBCK1	Pro98Arg	D	NA	NA	0	NA	NA
CHT00189	RBCK1	Arg130Leu	D	B	B	0.007	0.0002	0.0037
CHT00376	TAB3	Leu223Phe	D	D	P	0.0003	NA	NA
CHT00094	TIRAP	Lys84Glu	T	P	B	0.0002	NA	NA

CHT00160	TNFRSF10A	Leu432Trp	D	D	D	NA	NA	NA
CHT00160	TNFRSF10A	Gly60Arg	D	B	B	0.0011	3.3E-05	0.0006

Reviewer #3 (Remarks to the Author):

In this study, the authors identified a subtype of thyrocytes (i.e., TFC-1) that was located in the center of thyroid tissues of postnatal mice using 10X single-cell RNA-seq and spatial transcriptome sequencing technologies. This subtype of thyrocytes exhibited high migratory capacity by TNF- α -induced NF- κ B activation, which may drive thyroid folliculogenesis. The authors further demonstrated that TNF- α -induced NF- κ B activation of thyroid epithelial cells was triggered by myeloid cells in thyroid tissues, as shown by the cell-cell interaction analysis of single-cell data and experimental validation. In addition, they found that the dysregulation of the TNF- α -NF- κ B pathway was associated with the pathogenesis of thyroid dysgenesis (TD).

These datasets and findings will be an important resource for the field. However, it is a sub-optimally written paper and the omics data analysis is not comprehensive enough. Specific comments and concerns are as follows:

Major concerns:

1. Since the study is designed to investigate the mechanisms underlined transformation of thyroid primordium into mature follicles by dynamical observation of the process of thyroid follicle formation, the authors obtained single-cell datasets from the thyroid tissues of mice on postnatal 5 days, 10 days, 20 days, and 30 days. However, the developmental trajectory inference in single-cell data analyses is completely ignored. At least the authors should perform pseudotime analysis on thyroid follicular cells to assess the alterations of the transcriptome, the potential regulators, and the signaling pathways in thyroid follicular cell subsets during different postnatal days.

Response: Thank you for your good advice. We performed pseudotime analysis on thyroid follicular cells to assess the molecular difference in thyroid follicular cell subsets during different postnatal days and added these data in our revised manuscript. The results are shown as follows: “To verify the molecular changes of thyrocytes transition during thyroid development, we performed pseudotime analysis using Monocle2. Along the pseudotime trajectory, a gradually reduced faction of TFC-1 and increased TFC-2 from postnatal 5d to 30d were observed (Fig. S4a-4c). The result showed that the TFC-1 cells preferentially exist in the early stages of development, and its proportion gradually decreases over time, while the TFC-2 cells increased in the maturation stage of the thyroid. Genes increased in the TFC-1 branch (module 2) were largely involved in TNF- α -NF- κ B signaling and EMT transition, while those upregulated in TFC-2 (module 1) were enriched in oxidative phosphorylation (Fig. S4d-

4e).”

2. It is understandable that TFCs were selected for in-depth analysis. However, the information on the subpopulation distributions of a large number of cell types (~25000/30000) and their compositional changes at different postnatal timepoints was ignored. The authors should provide a paragraph in the main text with an overall description of the single-cell transcriptome profiling of thyroid tissue cells.

Response: Thanks for your suggestion. In the revised manuscript, we provided a paragraph describing the single-cell transcriptome profiling of thyroid tissue cells in the main text as follows: “To fully comprehend the thyrocytes heterogeneity, the mice thyroid tissues on different postnatal days (5d, 10d, 20d, and 30d) were dissected to perform single-cell RNA sequencing (scRNA-seq) (Fig. S2a). After standard data processing and quality control procedures (Methods), we obtained transcriptomic profiles for 29,914 cells (Fig. S2b-g). Further integration and unsupervised clustering revealed twelve major cell types in mice thyroid tissues (Fig. S2h). Annotation using canonical markers (Methods) in these 12 clusters identified them as T lymphocyte (T), macrophage (M), B lymphocyte (B), dendritic cell (DC), neutrophil (Neu), mastocyte (Mas), endothelial cell (EC), fibroblast (Fib), epithelial cell (Epi), stromal cell (Str), and lymphatic endothelial cell (LEC), as well as one group of nerve cells (Ner) (Fig. S2h-i).”

3. The description of findings in the analyses of scRNA-seq data is incoherent.

—Line 95-98. What is the point of this result supposed to infer? How does it relate to the above context, especially after adding the timepoint information?

Response: We are sorry for our incoherent description of the data. We adjusted the order of the figures in this section to make the presentation more logical and understandable in our revised manuscript. The new descriptions are as follows: “Among these, TFCs were selected for in-depth analysis (Fig. S3a-d). Support the gradual differentiation of TFCs during this time window, the expression levels of transcription factors involved in TFCs specification, such as *Pax8*, *Nkx2-1* and *Foxe1* were reduced, but the functions related genes such as *Duox2*, *Tg* and *Tpo* were increased in TFCs from day 5 to 1 month after birth (Fig. 2a). We also found that with thyroid development, the expression levels of genes related to cell adhesion gradually increased, while those related to cell mobility, such as *Vim*, *Mcam* and *Map4k4*, were gradually decreased (Fig. 2b). This was also confirmed by immunofluorescence and RNA-scope analysis (Fig. 2c-e).

The TFCs could be clustered into two subtypes, TFC-1 and TFC-2 by unsupervised clustering (Fig. 2f). With growth, the relative proportion of TFC-1 was reduced, with a

corresponding increased fraction of TFC-2 subtype (Fig. 2g). Interestingly, the genes related to positive regulation of cell migration were enriched in TFC-1 when compared with TFC-2 by gene ontology (GO) analysis (Fig. 2h and Fig. S3e). We further found that TNF- α -NF- κ B signaling pathway was significantly enriched in TFC-1 cells compared to the TFC-2 subtype according to the results of GO_KEGG signaling pathway analysis and gene set enrichment analysis (GSEA) (Fig. 2i,j and Fig. S3f). This indicates the activation of TNF- α -NF- κ B signaling and cell migration pathways in TFC-1 cells. Moreover, a gradual reduction of expression levels of TNF- α -NF- κ B transcripts in TFCs from postnatal 5d to 30d were observed (Fig. 2k).”

—Line 98-102. Based on the results that TNF- α -NF- κ B was significantly enriched in TFC-1 compared to the TFC-2 subtype, the authors could not obtain an inference of time-dependent activation of the TNF- α -NF- κ B signaling pathway.

Response: Thank you for the reviewer’s corrections. We have re-described the results of this section in our revised manuscript as follows: “We further found that TNF- α -NF- κ B signaling pathway was significantly enriched in TFC-1 cells compared to the TFC-2 subtype according to the results of GO_KEGG signaling pathway analysis and gene set enrichment analysis (GSEA) (Fig. 2i,j and Fig. S3f). This indicates the preferential activation of TNF- α -NF- κ B signaling and cell migration pathways in TFC-1 cells.”

—I am not sure how many similar incoherent issues exist elsewhere.

Response: We have examined the data and descriptions throughout the entire manuscript and made adjustments to address any potential incoherencies.

4. Related to the 1st comment, the authors obtained single-cell datasets from the thyroid tissues of mice on postnatal 5 days, 10 days, 20 days, and 30 days according to the study design. Why did the authors perform 10X spatial transcriptome sequencing on thyroid glands at postnatal 10 days only? It seems to me that the authors had no interest in investigating transcriptomic alterations and spatiotemporal dynamics in the thyroid tissues of distinct postnatal time points.

Response: We are sorry for not describing our rationale underlying 10 \times spatial transcriptome sequencing. It is based on our two findings: firstly, the histology and immunostaining results show that there seemed existed two types of thyrocytes, one already formed thyroid follicles located relatively in the peripheral of thyroid tissue, and another one type still not form follicles located in the central gland; secondly, the scRNA-seq analysis revealed there existed a subtype of thyrocytes with higher migratory potential and NF- κ B activation. We want to utilize the 10 \times spatial transcriptome sequencing to unbiasedly reveal whether the immature thyrocytes located in the central thyroid are, in fact, the thyrocyte with higher migratory and NF- κ B

activation identified by scRNA-seq. To make the overall rationale and coherence clear, we have included this comment before displaying the 10× spatial sequencing results.

5. Did the authors take the spatial context into account in their data analysis of cellular interactions? Several integration methods (e.g., Tangram (Biancalani et al. Nat. Methods 2021), Cell2location (Kleshchevnikov et al. Nat. Biotech. 2021), RCTD (Cable et al. Nat. Biotech. 2021)) can leverage cell type profiles learned from single-cell RNA-seq to decompose cell type mixtures in spatial transcriptomics, and estimate the spatial co-localization between distinct cell types, which can help facilitate cell-to-cell interaction analysis with spatial information.

Response: Decompose cell type mixtures in spatial spots is highly useful for downstream differential gene analysis and assessing cellular interaction. RCTD analysis (Cable et al. Nat. Biotech. 2021) was used to estimate the proportion in each pixel using cell-type annotated scRNA-seq reference. As shown in the scatterplots created by weights of different cell types obtained by RCTD, it seemed did not work very well on our data sets, as cell types were similarly distributed among spots (see below **a**). But using this method, we did observe a consistent correlation between TFC clusters leveraged by RCTD and unsupervised clustering analysis of spatial transcriptomic data (see below **b** and **c**).

Minor concerns:

1. I strongly recommend that the overall language of the main text should be polished with the assistance of professional English editors to improve the cogency and organization of the manuscript.

Response: Thanks for your suggestion. We have received assistance from a professional editor proficient in English to improve the cogency of the manuscript.

2. What doublet detection and removal method was used for the analyses of scRNA-seq data? This is not stated.

Response: Thanks for your scrutiny. We have added the doublet detection and removal method in our revised manuscript as follows “Computational inference of doublets was performed by the Scrublet package. The expected doublet rate was set to 0.06. Doublets identified in each sample individually were excluded from the following analyses.”

3. It will be useful to show the UMAP visualizations or clustering consistency and findings following batch correction using one more method (e.g., Harmony (Korsunsky et al., Nat. Methods 2019), LIGER (Welch et al. Cell 2019)) for the analyses of scRNA-seq data. Batch effects if not adequately addressed will affect findings and conclusions.

Response: Thanks for your suggestion. We used “RunHarmony” function in the harmony package to correct batch effect (Fig.S2k). After setting sample identity as the covariate, consistent clustering was visualized by UMAP, except the LEC (lymphatic endothelial cell), which was not separated from the EC (endothelial cell).

4. It is not appropriate to show different gene combinations in different figure panels (Fig. 2m, Fig. S3e, Fig. 3g) for the same TNF- α -NF- κ B signaling pathway.

Response: Thanks for your suggestion. Indeed, the same gene set of the TNF- α -NF- κ B signaling pathway obtained from the HALLMARK_TNFA_SIGNALING_VIA_NFKB database, which comprises 200 genes in total, was put for analyzing both the single cell sequencing and spatial sequencing data. However, spatial transcriptomic analysis can detect genes with lower expression levels, leading to more TNF- α -NF- κ B signaling pathway-related genes displayed than scRNA-seq data.

5. What did the results of the PCA indicate? Fibroblast and epithelial cells are not distinguished from each other in Fig. S2i.

Response: We are sorry for not describing it clearly. It shows how the cell types identified by scRNA-seq were plotted against the first two principal components of the

data. As this plot did not convey much information, we deleted it in the revised manuscript.

6. Line 87-90, Why are the changes in the expression of several transcription factors mentioned in the paper within the authors' expectations? At least, references should be added.

Response: Thanks for your scrutiny. We have modified the sentence into “Supporting the gradual differentiation of TFCs during this time window, the expression levels of transcription factors involved in TFCs specification, such as *Pax8*, *Nkx2-1* and *Foxe1* were reduced, but the functions related genes such as *Duox2*, *Tg* and *Tpo* were increased in TFCs from day 5 to 1 month after birth” and added the references in our revised manuscript.

7. Line 112, “Although the low resolution of ST-seq that one spot containing 50 cells made it difficult to distinguish single cells that were scattered in the thyroid tissue”, did the authors perform an analysis to assess the average cell number per spot in ST slice for thyroid gland?—When our group analyzed the average cell number per spot in our research system (different tissues from thyroid glands), one spot contained ~10-20 cells (10× Visium).

Response: Thank you for your question. We apologize for this imprecise description. To estimate the number of cells captured in a 10× spatial transcriptomics spot (55µm), we consider both the thickness of the tissue section and the size of thyroid cells. We initially roughly estimated that each spot might contain about 50 cells. However, the exact number of cells captured in each spot is unknown. We have revised our description of this result as follows: “Despite the low resolution of ST-seq, which makes it difficult to distinguish single cells scattered throughout the thyroid tissue, we observed significant overlaps between thyroid region-specific and cell type-specific genes, indicating the usefulness of employing MIA to infer cell types identified using scRNA-seq data on the thyroid region by integrating ST data”.

8. In Fig. 3c, I noticed that there are only 13 upregulated DEGs in TFC-2 cells compared with TFC-1 cells, while 2556 upregulated DEGs in TFC-1 cells compared with TFC-2 cells in scRNA-seq data. Please double-check the analyzed data and provide a supplementary table for the DEGs between TFC-1 cells and TFC-2 cells.

Response: Thanks for your suggestion. After rechecking the scRNA-seq data in Fig. 3c, we found that there were indeed 13 upregulated DEGs in TFC-2 cells compared with TFC-1 cells, while 2556 upregulated DEGs in TFC-1 cells compared with TFC-2

cells. The DEGs between TFC-1 cells and TFC-2 cells were provided in the Source data in our revised manuscript.

9. The panel number of Fig. 3 is not provided. There is a similar issue in Fig. 6.

Response: Thanks for your scrutiny. We have added the panel number of Fig. 3 and Fig. 6. in our revised manuscript.

Reviewer #4 (Remarks to the Author):

The authors analysed an impressive amount of data to describe a link of the TNF- α -NF- κ B pathway in development of the thyroid gland. With the developmental process being largely undescribed this manuscript provides important information. Unfortunately, the data is overwhelming and the presentation does not always help to digest the information. It would help if the authors reduce data in the main article and focus on the main points they want to convey with a better structure of the text.

A concern is that the authors might be looking at two different processes: migration of the whole gland to the correct anatomical position (eg in some Zebrafish data, <72hpf) and migration of individual follicles (in new born mice and older zebrafish) within the thyroid gland? While regulation might be involving similar pathways, it would make sense to pull these two processes apart?

Response: We absolutely agree with your opinion, and developmental time is critically important. Indeed, we are also interested in discriminating the molecular and biological differences between the migration of the whole gland and individual follicles. In zebrafish, the migration of the whole gland mainly occurs between 35-45hpf. At 55hpf, although the whole gland migration might continue, individual follicles separating from the whole primordium began dominant. The timing of the key morphological changes was also summarized in Table 2 of the review by Fagman, H. & Nilsson, M, which was attached below. (Morphogenesis of the thyroid gland. Mol Cell Endocrinol 323, 35-54 (2010)). Supporting this, our in vivo time-lapse tracing data in zebrafish also revealed that the critical time required for thyroid folliculogenesis lies between 50hpf to 5dpf (Fig. 1a).

We also realized that pulling these two processes apart in our models is very important. In order to check the early whole gland migration processes, we examined thyroid development in 50hpf in zebrafish embryos using WISH staining of thyroglobulin staining in this revised manuscript (data shown in revised Fig S9f, S9g and S12b). Additionally, we detailed and illustrated the thyroid development process in the introduction part of the revised manuscript.

Table 2
Timing of key morphogenetic events during thyroid development in different species.

	Specification	Budding	Migration	Follicle formation
Human ^a	E20–22	E24	E25–50	E70
Mouse ^b	E8.5	E10	E10.5–13.5	E15.5
Zebrafish ^c	24 hpf	32 hpf	35–45 hpf	55 hpf

E, embryonic day; hpf, hours post-fertilization.
^a De Felice and Di Lauro (2004).
^b Fagman et al. (2006).
^c Alt et al. (2006b).

The link with human recurrent biallelic variations of core genes in TNF- α -NF- κ B pathway is rather weak analysing tables “Table S9-S10”:

Prediction shows IKBKB and IKBKG variations are predicted benign, two carriers have no gland phenotype, two other either Thyroid dysgnesis or Autoinflammation.

CHT00217, IRAK1: with Thyroid dysgnesis , but no autoimmune, is predicted benign

CHT00189, RBCK1: prediction not uniform, Autoinflammation, with no thyroid phenotype.

Most references (35 to 40) cited in the discussion refer to autoimmune hypothyroidism, what can be considered a different disease mechanism, while they might influence the progress of the phenotype, they are not reported to be linked with migration/development described here.

Investigation of large data sets of whole gene sequencing with a description of the thyroid phenotype/autoimmune disease might be helpful.

Response: We truly appreciate your comments regarding the weak supporting data about the patients. As with the reason behind thyroid dysfunction, we think that continuous and detailed examinations of both the thyroid and immune functions among our CH newborns might be useful to discriminate the two. It’s a very good idea to describing the thyroid phenotype/autoimmune disease with large set of CH patients carrying all other genes mutations. However, as there are only a few patients with immune function data available, we could not get the description data now. We are working on recruiting back more patients and trying to assess the immune functions of them. At this stage, according to the suggestions of reviewers and editors, we removed the data and discussion about the patients in the revised manuscript. Hope our further investigations would lead us a better understanding between immune dysfunction and CH development.

In the discussion the authors mention that here are potential alternative described pathways (eg in some references in introduction) that could be involved, this can easily be investigated using eg RNA-seq data, other examples NIS can be negatively regulated by TNFalpha <https://pubmed.ncbi.nlm.nih.gov/32049985/>.

Was NIS expression assessed in their model?

Response: Thanks for your good advice. Using our scRNA-seq data, we compared the differential gene expression between TFC-1 and TFC-2. Indeed, alongside TNF- α -NF- κ B signaling, pathways like MAPK and TGF β were also upregulated in TFC-1, suggesting they might also be involved in thyroid folliculogenesis (Fig. 2i). We think that it might somewhat make sense that TNF- α negatively regulated NIS expression, as the function of which is involved in thyroid hormone synthesis. In our working model, TNF- α transiently promoted a migratory state in a subset of thyroid epithelial cells during a specific time window required for thyroid folliculogenesis. These migratory cells might need to maintain a relative immature state to migrate out of TP. The expression of NIS (*slc5a5*) was examined in our model by WISH. Lenalidomide treatment induced abnormally clumped *slc5a5* expression pattern in 5 dpf zebrafish, confirming the role of NF- κ B signaling in thyroid folliculogenesis (data shown in Revised Fig S12g).

In Ref 10 SOX9 shows, in an earlier stage of development, a similar expression pattern. Is there an acute (1-5 days?) effect on thyroid function with drug treatments used, this before thyroid follicle development is affected?

Response: As you suggested, to figure out whether TNF- α -NF- κ B signaling affected thyroid development before folliculogenesis, we examined thyroid development in 50hpf zebrafish embryos (before folliculogenesis in Zebrafish) using WISH staining of *thyroglobulin* with different drugs treatment. We discovered that lenalidomide, BMS and TPCA-1 could not affect thyroid development prior to follicle development (data shown in revised Fig S9f, S9g).

Absence of figure numbers on main text figure and changing colours of the targets eg fig 5b-c-d CD68 change, also in other figs, makes it harder to read. Not all data has been quantified, this would be sensible to come to a conclusion.

Response: Thanks for your kind suggestions. We made some adjustments to these figures so that the same target exhibits the same color, and numbers were added to make these easier to read. The quantification results were also included.

P4

Line56-58: “found folliculogenesis is not synchronous because peripheral thyrocytes formed a monolayer earlier than the central thyrocytes, which initially appeared as closed spheres (Fig. 1b).” This is not obvious from this picture, The TPO staining in Supplementary Fig. 1b are much more convincing, while the E-cadherin staining with the same days of maturation, does not show the same pattern. Quantification can clarify this.

Response: Thanks for your suggestion. The numbers of follicles with E-cadherin expression on the basolateral surface of epithelial cells were quantified in Fig. S1e.

Fig 4A suppl fig 6:

According to your ref 6 migration of TP is not finished until 72hpf, so 2.5 days and the 52-62.5hpf window fall well within that. GFP and mCherry are stable proteins (day-hours?) and might not be reflecting the correct response time of gene expression?

Response: As answered in your first question, the whole thyroid gland migration process generally finished around 45hpf in zebrafish. Your concern about whether the GFP signal truly reflect *in vivo* NF- κ B activation time is reasonable. Previous studies showed that the GFP fluorescence signal could be detected two hours after upstream NF- κ B stimulating (<https://pubmed.ncbi.nlm.nih.gov/26846883/>) and the GFP protein has a half-life of approximately 26 hours. In zebrafish, the GFP signal in thyroid primordium is barely detectable before 45hpf and gradually increased from 50hpf. So examination of the GFP activation in 2.5dpf and 52-62.5hpf zebrafish could possibly be used to reflect the GFP response time *in vivo*.

P7

line 160 “After treatment with TPCA1, thyroid function was compromised”

What is the thyroid hormone status in the zebrafish? Is there an acute (1-5 days?) effect on thyroid function after different drug treatment?

Response: We examined the thyroid hormone status in zebrafish using immunofluorescence staining of thyroxine (T4). Along with defective thyroid folliculogenesis, TPCA1 treatment also led to reduced generation of thyroxine producing follicles in zebrafish at 8dpf (shown in revised Fig S9b). However, in 50dpf zebrafish, the thyroid hormone level is very low and beyond the sensitivity of antibody detection. Instead, we show that neither BMS nor TPCA-1 could influence thyroid development before folliculogenesis by WISH staining of *thyroglobulin* (shown in revised Fig S9f, S9g).

TPCA-1 is a direct dual inhibitor of STAT3 and NF- κ B, did the authors look at the effect on STAT3 related pathways?

Response: Yes, your concern is truly worthy of paying attention to. Nonspecific effects often accompany the small molecule inhibitors, which is also why we have chosen two drugs, TPCA-1 and BMS-345541, to study the influence of NF- κ B signaling. We also overexpressed the active form of NFKBIA in thyroid cells to prove the specific role of NF- κ B signaling involved. The influence of TPCA-1 on STAT3 signaling and thyroid function is an interesting topic and will need further investigation.

P8 line 172-176

“We further specifically overexpressed the constitutively active form of NFKBIA (I κ B α) in thyrocytes under the thyroglobulin promoter, which retains NF- κ B members in the cytosol to resist NF- κ B activation. Transient overexpression of this plasmid in thyrocytes induced similar thyroid abnormalities, confirming the intrinsic role of NF- κ B signaling in thyroid development (Supplementary Fig. 8e,f).”

Did the authors show in this model similar outgrowth. No details about transfection, methods – efficiency - ...

explain contradiction in Supplementary Fig. 8e,

“Representative images showing thyroid specific inhibition of NF- κ B activation by constitutively overexpression nfkbia under thyroglobulin promoter”

“Transient overexpression of this plasmid” in text versus “constitutively overexpression” in Supplementary Fig. 8e,

P8

Line 174-5“Transient overexpression of this plasmid in thyrocytes induced similar thyroid abnormalities”

I did not find the Transient overexpressing method in zebrafish.

Response: We are very sorry about our confusing description. In this paper, all experiments inhibiting NF- κ B activation, specifically in thyroid cells of zebrafish, were done by transient overexpression. In using the word “constitutive”, we want to convey that the used mutated form of NFKBIA plasmid (S32/36A mutation) can suppress NF- κ B activation constitutively. We have modified the statement in the revised manuscript and added a detailed description of the revised materials and methods.

P8

“184 Myeloid cells in the developing thyroid tissue secrete TNF- α to trigger the NF- κ B 185 activation of TFCs”

Refer to species used in the beginning of this section

Supplementary Fig. 9: Refer to species used

Response: Thanks for your suggestion, we have added species in this section.

P9, line 206-213

The Authors need to take care comparing events in zebrafish at 2dpf and 5dpf. At 2dpf the TP is still migrating to its final position in the zebrafish, according to ref 6. At day 5dpf it is more likely a maturation process, similar to what is seen in mice.

Response: We totally understand your concern. According to the thyroid development timeline shown in the answering to your first question, we discriminated the TP whole gland migration process from the later maturation process by WISH staining of *thyroglobulin* expression in 50hpf zebrafish. We also carefully checked all the conclusions we got related with folliculogenesis were between the time critical for thyroid follicle formation.

fig 5b-c-d CD68 does not surround p65 positive cells but might be in close proximity, fig 5c, suggest the other way round. Quantification?

Response: Yes, we changed it to in close proximity in the revised version. The quantification results were also added in this revised version.

P10-11,

“248 Recurrent biallelic variations of core genes in TNF- α -NF- κ B pathway identified

249 from CH patients with an unexplained etiology” Can we assume that genes from Fig2 - fig 3 not appearing in table S7 did not show relevant variations?

Response: We are sorry for our unclear descriptions. In analyzing TNF- α -NF- κ B pathway mutations in CH patients, we only chosen 38 genes, all are well-known and have relative specific roles on TNF- α -NF- κ B signaling pathway. However, in analyzing the differential expressed genes among TFC-1 and TFC-2 in mouse thyroid tissues, 200 genes obtained from HALLMARK_TNFA_SIGNALING_VIA_NFKB database were put into analysis. These 200 genes have more diversified role and some of which may not directly involved in TNF- α -NF- κ B signaling pathway. Thus, we only examined the 38 core TNF- α -NF- κ B signaling pathway genes mutations in CH patients.

If I understand correctly only 4 individuals from 168 with TNF- α -NF- κ B pathway link have also an autoimmune disease?

Response: Sorry for our unclear description. Indeed, only a small subset of CH patients with data regarding immune function are available. Most of them remain unexamined. Of the 13 patients carrying biallelic variations of core TNF- α -NF- κ B pathway genes, only eight patients with immune function were examined. Furthermore, 4 of the eight individuals were found to have immune dysfunction.

What is the incidence of autoimmune disease in the 113/281 patients with no TNF- α -NF- κ B pathway link?

Response: It is a very good idea to check the immune function in CH patients without TNF- α -NF- κ B pathway gene mutations. We are working on describing this incidence by recruiting back these patients and getting immune function examinations on them. We hope our further studies would get a clear understanding of the relationships between thyroid and immune dysfunction in patients.

None of these genes (except NFKB1) appear in the TNF- α -NF- κ B regulated genes in fig 2m, fig3g, how are they affected?

Response: If combining all the 200 genes from HALLMARK_TNFA_SIGNALING_VIA_NFKB database, mutations of 147 genes were found to occurred in 268 of 281 CH patients with undefined etiology.

REVIEWER COMMENTS

Reviewer #1 (Remarks to the Author):

The Authors provide experimental data in vitro and in vivo supporting the relevant role played by myeloid cells in the process of thyroid folliculogenesis.

In particular, TNF- α deriving from myeloid cells activates NF-kB which in turns would promote the migration of thyrocytes for follicle generation. These evidence were obtained in two vertebrate models, mice and zebrafish, where adult thyroid structure is variably organized. These data point to a relevant role of this pathway in the early stages of folliculogenesis. The translational value of the manuscript is reduced after the removal of human's data, but this revised version is probably more solid than previous one as several doubts were raised by the the benign prediction of identified variants. However, the whole of these results lead me to an alternative interpretation of the model of hypothyroidism here presented.

Comments:

- The thyroid structure of TPCA1 mice shows a decreased number of follicles of large size, which is indicative of a possible compensation mechanism, perhaps induced by the raised TSH levels (which in fact is significant but not indicative of severe primary hypothyroidism). If the TPCA1 treatment is withdrawn at a certain growth stage these figures may be consistent with the idea that this may represent a model of colloid goitrogenesis. A transient immune/inflammatory interference might also occur during the early development in fetuses and neonates of mothers with autoimmune thyroiditis, and the models here presented be consistent with transient forms of hypothyroidism evolving later in colloid goiters. Such idea would then be consistent with the absence of thyroid autoimmune markers in neonates with CH. I would then find appropriate to study thyroid size and TFT in adult mice at later time points after a short early course of TPCA1 treatment.
- To evaluate the entity of thyroid function alteration during development I would also find appropriate to show TSH β expression, in addition to T4, in cebpa mutant zebrafish, but in both embryos and adults.

Reviewer #2 (Remarks to the Author):

The authors responded adequately to all the questions, and followed most of the suggestion. I agree to publish this work.

Reviewer #3 (Remarks to the Author):

The authors have made substantial revisions, including adding pseudotime analysis, a description of single-cell transcriptome profiling, improved coherence in result descriptions, and clarification of their rationale for spatial transcriptome sequencing. Overall, these revisions have basically addressed the comments raised and made appropriate revisions to improve the clarity and comprehensiveness of their study.

Reviewer #4 (Remarks to the Author):

The Authors produce evidence in animal models (zebrafish and mice) of the interaction of myeloid cells and TFC during thyroid morphogenesis, with the TNF α - NF- κ B pathway involved in development of the thyroid gland. Different thyrocytes were identified, TFC1, (possible progenitor cells), and TFC2, mature thyroid cells that tend to be functional is meaningful.

The data are original and provide new insights into the mechanisms of thyroid follicle growth. The results are obtained with up-to-date technologies and their illustrations are clear.

The authors compiled adequate answers to reviewers questions and the overall quality of the manuscript has significant improved.

A few minor questions:

The abbreviation TFC appears first on line 121, with no explanation where it stands for, this not is mentioned until line 247. Please update.

Supplemental figure S2h. The Major cell types identified from thyroid tissues are denoted. According to the shape I denote that the PTC /C-cell/ TFC are in the epithelial cells cluster. Are there non- PTC /C-cell/ TFC in this cluster, if so can you distinguish them?

I the main text could you (line 121) point out that they are in the epithelial cells cluster.

This might just show my ignorance on this analysis, but Supplemental figure S2h versus S2K, the epithelial cells cluster looks different thus this suggest overlapping between clusters? Are there any cells in the other clusters with thyroid markers? How are the TFC1 and 2 organised in that analysis?

Line 152-156: I like the summary, but please adjust this sentence.

NF- κ B suppression during development affects thyroid, how do the inhibitors used affect thyroid function in adult animals, once the thyroid is fully matured? If compounds used only affect development only, it should have only a minor effect on mature thyroid gland functioning, or maybe affect regeneration/maintenance long term? If you could refer to published literature would be sufficient.

Fig. S13. Tnf- α suppression on vascular development in zebrafish. This figure also contains human (d. Nthy-Ori3) and mouse (e) data please adjust title.

REVIEWER COMMENTS

Reviewer #1 (Remarks to the Author):

The Authors provide experimental data *in vitro* and *in vivo* supporting the relevant role played by myeloid cells in the process of thyroid folliculogenesis.

In particular, TNF- α deriving from myeloid cells activates NF- κ B which in turns would promote the migration of thyrocytes for follicle generation. These evidence were obtained in two vertebrate models, mice and zebrafish, where adult thyroid structure is variably organized. These data point to a relevant role of this pathway in the early stages of folliculogenesis. The translational value of the manuscript is reduced after the removal of human's data, but this revised version is probably more solid than previous one as several doubts were raised by the the benign prediction of identified variants. However, the whole of these results lead me to an alternative interpretation of the model of hypothyroidism here presented.

Comments:

- The thyroid structure of TPCA1 mice shows a decreased number of follicles of large size, which is indicative of a possible compensation mechanism, perhaps induced by the raised TSH levels (which in fact is significant but not indicative of severe primary hypothyroidism). If the TPCA1 treatment is withdrawn at a certain growth stage these figures may be consistent with the idea that this may represent a model of colloid goitrogenesis. A transient immune/inflammatory interference might also occur during the early development in fetuses and neonates of mothers with autoimmune thyroiditis, and the models here presented be consistent with transient forms of hypothyroidism evolving later in colloid goiters. Such idea would then be consistent with the absence of thyroid autoimmune markers in neonates with CH. I would then find appropriate to study

thyroid size and TFT in adult mice at later time points after a short early course of TPCA1 treatment.

The disease model of transient hypothyroidism developed by defective TNF α -NF/ κ B signaling pathway proposed by the reviewer is intriguing. Previous studies have also supported a compensatory thyroid hyperplasia revealed as colloid goiters after birth could somewhat compensate thyroid dysfunction. Our current research findings indicate that the activation of the TNF α -NF/ κ B signaling pathway plays a crucial role in the follicular formation during thyroid development. Blockade and deficiency of this pathway during the early stages (within the first month after birth in mice) can lead to impaired follicular maturation and result in hypothyroidism. As TNF α inhibitory monoclonal antibody treatment of adult rheumatoid arthritis patients did not significantly alter thyroid function, suggesting that this signaling pathway may not be pivotal for proliferation and maintenance of thyroid follicles in adults. Therefore, in TPCA-1 treated mouse, if thyroid insufficiency could be partially compensated by thyroid cell proliferation later, it would probably create a model similar with transient congenital hypothyroidism in humans.

In fact, there are also clinical cases of congenital hypothyroidism that can restore thyroid function through excessive compensatory proliferation of thyroid follicular cells postnatally. For instance, in patients of congenital hypothyroidism resulting from biallelic mutations in *DUOX2* (our unpublished data) or *TPO* genes (PMID: 32088313). In general, these patients should have a high residual enzyme activity, so an increase in the number of thyroid cells will compensate for the deficiency in thyroid hormone synthesis caused by the deficiency in enzyme activity.

However, it is notable that embryonic thyroid tissue may have increased lymphocyte infiltration and greater NF- κ B activation when mother-associated autoimmune thyroid disorders are present. This would, in line with our theory, promote the development of thyroid follicular cells. In fact, it is hypothesized that transient hypothyroidism in offsprings of mothers with autoimmune thyroid disorders may be caused by thyroid autoantibodies entering the fetal body and mediating thyroid cell death (antibody-

mediated cytotoxicity, such as TGAb or TPOAb) or inhibiting thyroid hormone synthesis (e.g., TSHR inhibitory antibodies), instead of abnormal thyroid development.

We are attempting to explore adult thyroid function using the approach as the reviewer suggested, a short early exposure in neonate mice and examine thyroid function at later times. But as the revision time limits and we do not know how long would be needed for the colloid goiters to develop and thyroid dysfunction could be compensated in these TPCA1 treated mice, we are very sorry that we could not provide a satisfactory answer to your question now. But of course, you provided a fascinating hypothesis for us to investigate the data regarding CH patients carrying NF- κ B mutations in the future.

- To evaluate the entity of thyroid function alteration during development I would also find appropriate to show TSHba expression, in addition to T4, in *cebpa* mutant zebrafish, but in both embryos and adults.

Thank you for this suggestion. As the *cebpa* mutants could not survive beyond 2 weeks, we examined the *Tshba* in 6dpf WT and *cebpa* mutant zebrafish by WISH. Elevated *Tshba* were found in *cebpa* mutants as compared with sibling controls (Figure S12a).

Reviewer #2 (Remarks to the Author):

The authors responded adequately to all the questions, and followed most of the suggestion. I agree to publish this work.

Reviewer #3 (Remarks to the Author):

The authors have made substantial revisions, including adding pseudotime analysis, a description of single-cell transcriptome profiling, improved coherence in result descriptions, and clarification of their rationale for spatial transcriptome sequencing. Overall, these revisions have basically addressed the comments raised and made appropriate revisions to improve the clarity and comprehensiveness of their study.

Reviewer #4 (Remarks to the Author):

The Authors produce evidence in animal models (zebrafish and mice) of the interaction of myeloid cells and TFC during thyroid morphogenesis, with the TNF α - NF- κ B pathway involved in development of the thyroid gland. Different thyrocytes were identified, TFC1, (possible progenitor cells), and TFC2, mature thyroid cells that tend to be functional is meaningful.

The data are original and provide new insights into the mechanisms of thyroid follicle growth. The results are obtained with up-to-date technologies and their illustrations are clear.

The authors compiled adequate answers to reviewers questions and the overall quality of the manuscript has significant improved.

A few minor questions:

The abbreviation TFC appears first on line 121, with no explanation where it stands for, this not is mentioned until line 247. Please update.

Thank you for your kind reminder, we updated this in the revised manuscript.

Supplemental figure S2h. The Major cell types identified from thyroid tissues are denoted. According to the shape I denote that the PTC /C-cell/ TFC are in the epithelial cells cluster. Are there non- PTC /C-cell/ TFC in this cluster, if so can you distinguish them?

I the main text could you (line 121) point out that they are in the epithelial cells cluster.

We are sorry for the unclear description. The epithelial cluster shown in Supplemental figure S2h could be further classified into three subsets including PTC, C-cell and TFC, using markers including Calca, Pth and Tpo (Supplemental figure S3a and S3d). Only

the TFC were then selected for downstream analysis. We also pointed out this classification method in the revised manuscript.

This might just show my ignorance on this analysis, but Supplemental figure S2h versus S2K, the epithelial cells cluster looks different thus this suggest overlapping between clusters? Are there any cells in the other clusters with thyroid markers? How are the TFC1 and 2 organised in that analysis?

We confirmed our clustering consistency following batch correction using two different methods including Canonical correlation analysis (CCA) (Supplemental figure S2h) and Harmony (Supplemental figure S2k). As shown below Figure R1, the clusters generated by the two methods were consistently overlapped. The Epi cluster generated by CCA were used for downstream analysis in this study. As shown below Figure R2, thyroid markers like *Tg*, *Nkx2-1*, *Foxe1*, *Pax8*, *Tpo*, *Iyd* and *Dio1* were exclusively expressed Epi cluster in Harmony method. In addition, the TFC classified by Harmony can also be divided further into TFC-1 and TFC-2, which were consistently overlapped with CCA methods (Figure R3).

Figure R1. Analysis of cell cluster overlapping ratio between CCA and Harmony methods

Figure R2. The expression of the thyroid marker genes in the 11 cell clusters divided by the Harmony method

Figure R3. Analysis of thyroid cell cluster overlapping ratio between CCA and Harmony methods

Line 152-156: I like the summary, but please adjust this sentence.

Thank you. We adjusted this sentence to make it more confluent.

NF- κ B suppression during development affects thyroid, how do the inhibitors used affect thyroid function in adult animals, once the thyroid is fully matured? If compounds used only affect development only, it should have only a minor effect on mature thyroid gland functioning, or maybe affect regeneration/maintenance long term? If you could refer to published literature would be sufficient.

According to our dynamic tissue staining of mice thyroid tissues from birth to one month old, either follicle number or matured thyroxine follicles continued increased during this time. In fact, TNF α inhibitory monoclonal antibody treatment of adult rheumatoid arthritis patients did not significantly alter thyroid function, suggesting that this signaling pathway may not be pivotal for proliferation and maintenance of thyroid follicles in adults (PMID:28824542). We have also included this in the discussion part.

Fig. S13. Tnf- α suppression on vascular development in zebrafish. This figure also contains human (d. Nthy-Ori3) and mouse (e) data please adjust title.

Thanks, we have adjusted the title.

REVIEWERS' COMMENTS

Reviewer #1 (Remarks to the Author):

I wish to thank the Authors for the comments, they responded adequately also to this second revision round.